# Photocascade chemoselective controlling of ambident thio(seleno)cyanates with alkenes via catalyst modulation

Injamam Ul Hoque[1], Apurba Samanta[1], Shyamal Pramanik[1], Soumyadeep Roy Chowdhury[1], Rabindranath Lo[2] & Soumitra Maity [1] ✉

Controlling the ambient reactivity of thiocyanates in reaction manifolds has been a long-standing and formidable challenge. We report herein a photo-redox strategy for installing thiocyanates and isothiocyanates in a controlled chemoselective fashion by manipulating the ambient-SCN through catalyst modulation. The methodology allows redox-, and pot-economical 'on-demand' direct access to both hydrothiophene and pyrrolidine heterocycles from the same feedstock alkenes and bifunctional thiocyanomalonates in a photocascade sequence. Its excellent chemoselectivity profile was further expanded to access Se- and N-heterocycles by harnessing selenonitriles. Redox capability of the catalysts, which dictates the substrates to participate in a single or cascade catalytic cycle, was proposed as the key to the present chemodivergency of this process. In addition, detailed mechanistic insights are provided by a conjugation of extensive control experiments and dispersion-corrected density functional theory (DFT) calculations.

Control of chemoselectivity in the reaction involving ambient species is a long-standing issue in chemistry, and continuous efforts are being made to address its subtle nature in organic synthesis[1]. The thiocyanate ion is one such moiety which, if successfully tamed, can selectively lead to organic thiocyanates and isothiocyanates, both having a vast presence in bioactive molecules as well as serving as valuable synthetic intermediates (Fig. 1A)[2–6]. However, its chemoselectivity is dependent upon numerous factors, making it a perennially tricky problem to solve[1,7]. Though alkyl isothiocyanates are more stable than their thio-cyanate counterparts in thermodynamic terms, the alkylation of SCN⁻ usually leads to bonding through the S-center due to its kinetic pre-ference over N-attack in both $S_N2$ and $S_N1$ reactions (Fig. 1B)[8]. Tradi-tionally, alkyl thiocyanates have been accessed by subjecting alkyl halides to nucleophilic substitution by SCN⁻. In recent years, installa-tion of thiocyanates onto readily available feedstock alkenes is also quickly coming to the fore as a route of choice. This has commonly been achieved by three-component reactions, either involving the addition of thiocyanate radical (generated by oxidation of thiocyanate salts) onto an olefinic π−bond or trapping of thiocyanate anion by a

carbocationic intermediate, generated in the radical reactions of alkenes (Fig. 1C)[3,9]. Isothiocyanation of alkenes, on the other hand, is quite rare[10,11] making direct access to organic isothiocyanates limited to reactions of primary amines with electrophilic thiophosgene or carbon disulfide under harsh reaction conditions[12,13]. Hence, the rational design of a chemodivergent reaction manifold for selective incorporation of thiocyanate and isothiocyanate functionalities from the same reagent under mild conditions is highly desirable.

Chemodivergent strategies that allow chemists to access struc-turally diverse products from a common set of starting materials are highly valuable[14]. Recently, the advent of photoredox catalysis has further matured this strategy to access divergent products under mild conditions[15,16]. Unlike others, photocatalysts have the unique ability to promote reactions via both single-electron-transfer (SET)[17,18] and energy-transfer (EnT)[19] processes where redox potential and triplet energy level of the catalyst dictate their actions, respectively. Tactically merging the above two processes serially or by repeating either of the processes sequentially, a photo-cascade platform could be devised to possibly harness molecular complexity in a step- and redox-economic

[1]Department of Chemistry and Chemical Biology, Indian Institute of Technology (Indian School of Mines), Dhanbad, JH 826004, India. [2]Institute of Organic Chemistry and Biochemistry, Czech Academy of Sciences, Flemingovo náměstí 542/2, Prague 160 000, Czech Republic. ✉e-mail: smaity@iitism.ac.in

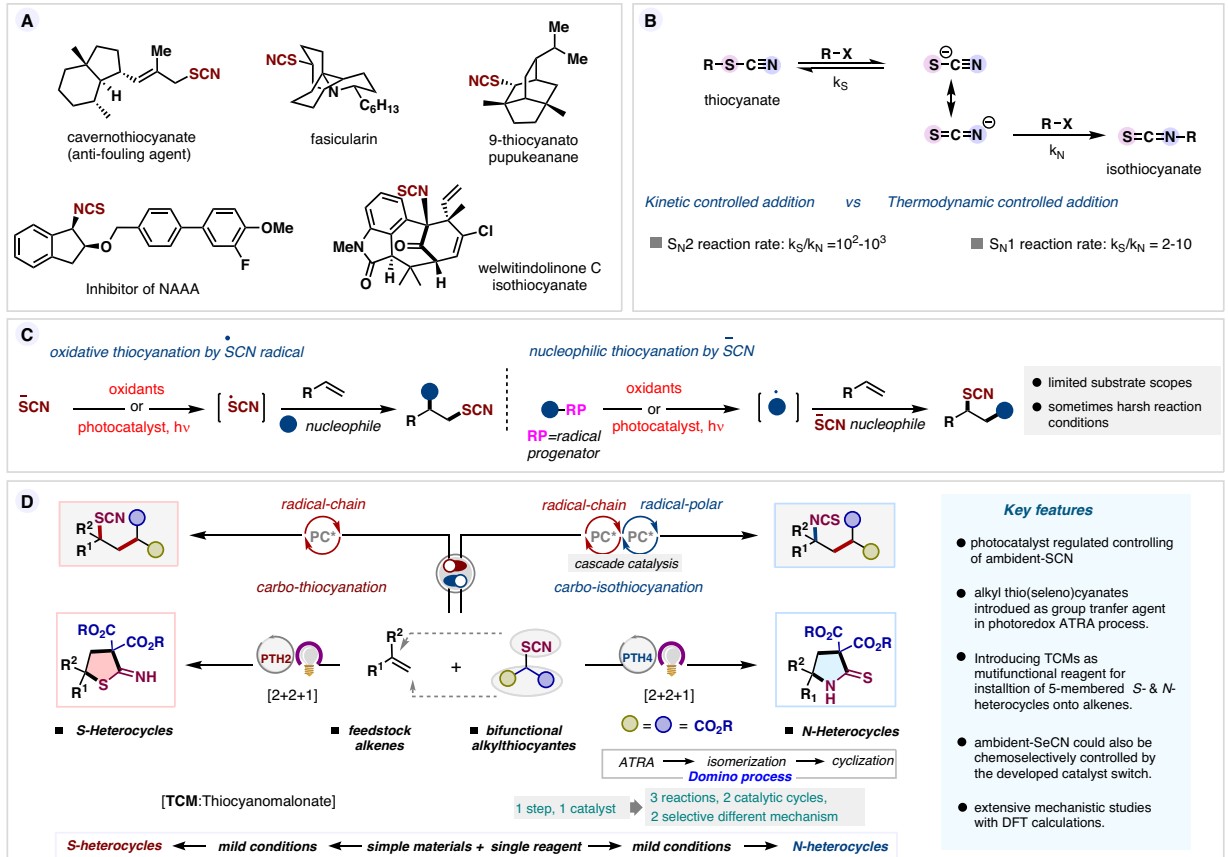

**Fig. 1 | Ambient reactivity of thiocyanate and present work. A** Biologically relevant thiocyanates and isothiocyanates. **B** Ambient reactivity profile of thiocyanate anion. **C** Reported three-component thiocyanation of alkenes. **D** This work: Catalyst-regulated thiocyanation and isothiocyanation by bi-functional alkylthiocyanate reagents.

way. In contrast to commonly encountered single-cycle catalytic platforms, multicycle cascade photocatalysis allows different reaction outcomes by sequentially engaging the first cycle product as an intermediate for the subsequent cycle. To be operative, this necessarily requires that the intermediate responds in any of the catalyst activation modes. Seminal works from Gilmour[20], Glorius[21], König[22], and others[23–25] have powerfully demonstrated this elegant strategy for achieving unprecedented transformations. We envisioned exploring this elegant platform for the controlled chemoselective incorporation of thiocyanate and isothiocyanate, en route to 5-membered *S*- and *N*-heterocycles from alkenes and thiocyanomalonates (TCMs) by a tactical regulation of the photocatalyst (Fig. 1D). Highly substituted 5-membered heterocyclic scaffolds are usually accessed by Lewis acid catalyzed [3 + 2] cycloaddition of donor-acceptor cyclopropanes (DACs) with dipolarophiles[26]. While this strategy is efficient, the requirement of using DACs has impeded their broad applications, particularly for late-stage functionalization of complex molecules. The use of olefins, which are widely accessible and inexpensive feedstock, in the place of DACs to afford 5-membered heterocycles in a step- and redox-economical way under mild conditions provides a challenging yet appealing solution. The proposed radical route involving TCMs would enable the use of unactivated olefins and thus have the potential to overcome the limitations of existing ionic methods.

We hypothesized that redox-neutral carbothiocyanation of alkene **1** to the difunctionalized product **3'** may be achieved by photo-reductive cleavage of thiocyanomalonate **2** followed by SCN-transfer radical addition (Fig. 2A)[27]. Coupling this ATRA process with a cyclization event may further allow access to 2-imino-tetrahydrothiophene (2-ITHT) product **3**. On the contrary, subjecting **3'** to a consecutive SET

event rather than cyclization can possibly lead to an oxidative radical-polar crossover[28] in a subsequent step which may allow thiocyanate **3'** to isomerize to the thermodynamically stable isothiocyanate **4'**[29–32]. *N*-cyclized 2-thiopyrrolidone **4** may then be readily accessed by a similar cyclization step as before. Splitting the overall reaction draft into ATRA and isomerization realms, a strategic key step was the judicious choice of photocatalyst which would selectively allow the alkyl thiocyanates (**2** and **3'**) to participate in a single or cascade catalytic cycles by matching its redox potentials. It is worth mentioning that the radical reactivity of alkyl thiocyanates, particularly as a thiocyanate group transfer agent, has never been explored[33]. However, such a road map for controlling ambient functionality would lead to alkylchalcogen-onitriles as multifunctional reagents, thus amplifying their synthetic utility. Continuing our research efforts on photoredox catalysis[34,35], we describe here the implementation of this blueprint enabling the chemodivergent synthesis of 5-membered *S*-, *Se*- and *N*- heterocycles from alkenes and thio(seleno)-cyano malonates, which highlights both the potential as well as the challenges of ambient thio(seleno)cyanates in organic synthesis.

## Results and Discussion

### Optimization of the reaction

Pursuing this idea, we set out to introduce thiocyanomalonates (TCMs) as bifunctional reagents[36] since some of the malonate congeners (-SePh, -TEMPO, -halo) have previously been exploited in radical reactions[27,37–39]. Based on the reduction potential of the model thiocyanates **2a** ($E_{1/2} = -1.32$ V vs. SCE) and **3a'** ($E_{1/2} = -1.81$ V vs. SCE), we anticipated that metal-free phenothiazine catalysts (PTH = *N*-arylphenothiazine) could deliver our objectives, courtesy of their superior and

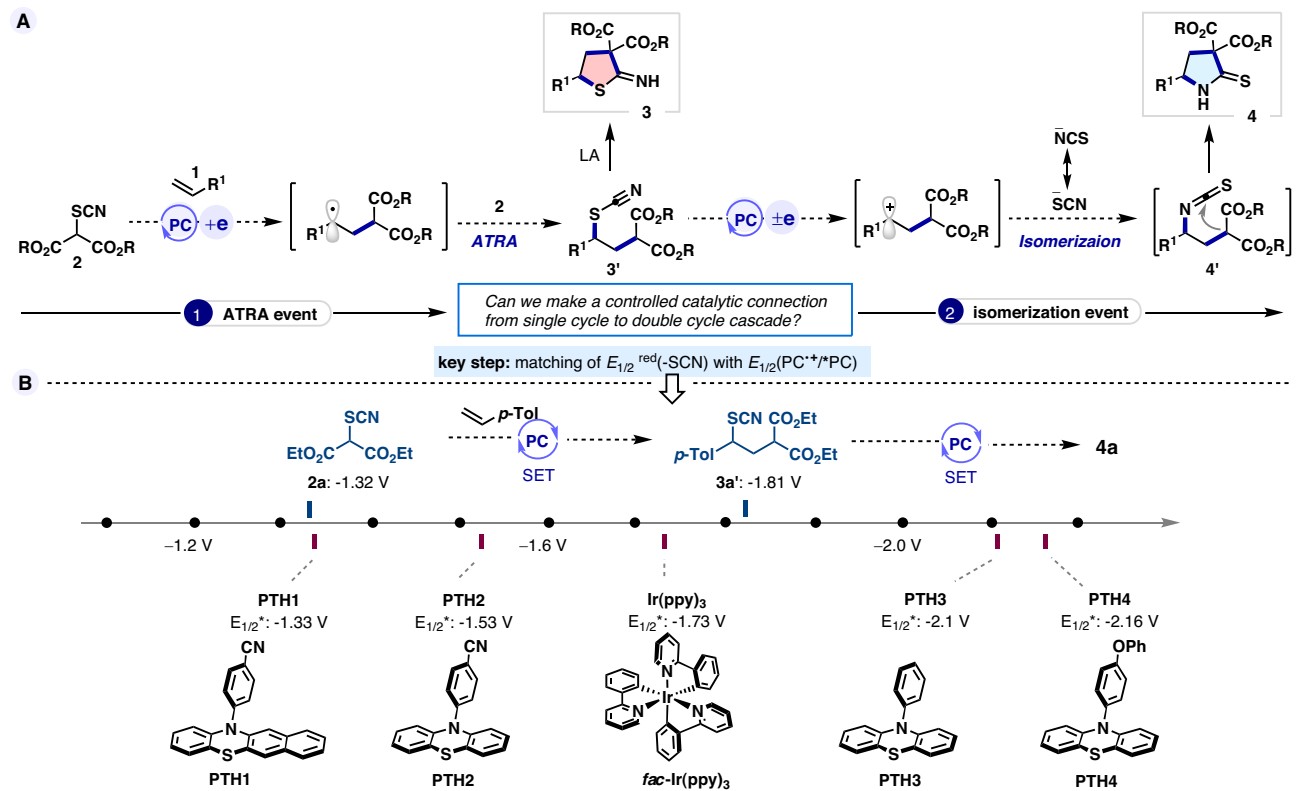

**Fig. 2 | Reaction design and catalyst evaluation. A** Working hypothesis. **B** Electrochemical scale (vs. SCE).

tunable redox capacity in the excited state (Fig. 2B)[40,41]. After screening the reaction parameters based on our working hypothesis (see Supplementary Table 1 for detail optimization), we found that the reaction between 4-methylstyrene **1a** and diethyl thiocyanomalonate **2a** occurred in the presence of **PTH1** under LED irradiation ($\lambda_{max}$ = 390 nm) in toluene to produce metastable ATRA product **3a′**, which during silica gel column chromatography or upon treatment with AlCl$_3$ in the same pot (just after the photoredox reaction) afforded the desired hydro thiophene product **3a** in 38% yield (entry 1, Table 1). The pursuit of higher yield led us to the synthesis of **PTH2**, which delightfully furnished **3a** in 89% yield (entry 2, Table 1). Further attempts at optimization by changing the catalyst (entries 3–5, Table 1) and solvent did not improve the yield of **3a** (entries 6–9, Table 1). The *S*-selective catalyst **PTH2** features a 4-CN substituent on its *N*-phenyl ring while its replacement with 4-OPh (**PTH4**) results in *N*-selectivity. It is to be noted that while the *S*-cyclization sequence needs assistance from Lewis acid, the *N*-cyclization variant i.e., 2-thiopyrrolidone **4a** (81%), is obtained as a direct product just under photoredox conditions through a spontaneous domino difunctionalization-isomerization-cyclization sequence (entry 10, Table 1). **3a** was obtained as the major product using commercially available **PTH3** in a short reaction time, which got converted exclusively to **4a** in the long run (entry 3 vs 11, Table 1). Other *S*-selective catalysts did not exhibit the same kind of chemodivergent isomerization in long reaction time (entries 12-13, Table 1). Overall, catalyst screening indicated that the electron-deficient *N*-aryl ring of PTH favored the *S*-heterocycle product, whereas unsubstituted or electron-rich *N*-aryl of PTH preferred *N*-heterocycle formation. These observations can be rationalized with their increasing reduction profile in the exited state[40], as desired to initiate the 2$^{nd}$ cycle for isomerization. Control studies revealed the necessity of catalyst and light for the reaction to occur (entries 14-15 and 16, Table 1)[42]. Aerial oxygen has a negative impact on the reaction yield (entry 17, Table 1). Additional control experiments ruled out the impact of thermal heating on the success of the reaction (entries 18-19,

Table 1). Finally, catalysts **PTH2** and **PTH4** were selected for accessing *S*-heterocycles and *N*-heterocycles respectively.

## Substrate scope

We examined the generality of hydrothiophene construction by exploring a variety of alkenes under conditions A (Fig. 3). Aromatic alkenes with different electronic properties or steric hindrance performed well, leading to 2-ITHT derivatives (**3a-k**) in good to excellent yields (58%–88%). Vinyl heteroarenes bearing furan, thiophene, indole, and thiazole rings were suitable substrates delivering corresponding products (**3l-o**), although the yield of the latter two is poor (35%–38%) potentially due to the decomposition of these alkenes in our reaction conditions. Substituted-styrenes (both *α*- and *β*-) also survived to provide the desired products (**3p-r**) satisfactorily (58%–84% yield). Unfortunately, triphenylethylene and stilbene were unreactive, likely because of steric hindrance. Similarly, alkynes were found to be incompatible to this reaction, forming complex and insoluble mixtures upon reaction with the TCMs (see the list of incompatible substrates, Supplementary Table 6). Next, we turned our attention to the more challenging unactivated alkenes, which have proven to be difficult substrates under the radical carbothiocyanation process[35,43]. Remarkably, this method can be extended to aliphatic alkenes. A range of terminal substituted aliphatic alkenes bearing various functionalities such as ester, ketone, alcohol, bromide, and sulfonate were well tolerated (**3s-x**) with good to excellent yield (63%–84%). Both *α*- (**3y-aa**) and *β*- (**3ab,ac**) substituted alkenes were also suitable substrates in this transformation (53%–72% yield). Surprisingly, electron-deficient acrylates (unsubstituted and *α*-substituted) provided the desired products (**3ad-ag**) as well, albeit obtained in modest yields (39%–47%), which has interesting mechanistic implications[44]. The ethyl group of **3a** could be substituted by other alkyl groups by choosing the appropriate TCM reagents (**3ah-ak**). Finally, the suitability of this mild photocatalytic method to pharmaceutically relevant molecules was demonstrated by derivatization of estrone (**3al**), naproxen (**3am**), oxaprozin (**3an**) as

**Table 1 | Optimization of the photochemical chemodivergent reaction[a]**

| Entry | Catalyst | Solvent | Time | 3a (%)[c] | 4a (%)[c] |
|---|---|---|---|---|---|
| 1 | PTH1 | Toluene | 1 h | 38 | 2 |
| 2 | PTH2 | Toluene | 30 min | 89 (86)[d] | 7 |
| 3 | PTH3 | Toluene | 20 min | 68 | 21 |
| 4 | PTH4 | Toluene | 20 min | 70 | 22 |
| 5[e] | $fac$-Ir(ppy)$_3$ | Toluene | 6 h | 67 | 8 |
| 6 | PTH2 | CH$_3$CN | 45 min | 73 | 6 |
| 7 | PTH2 | DMF | 1 h | 38 | 2 |
| 8 | PTH2 | DMSO | 1 h | 35 | 2 |
| 9 | PTH2 | 1,2-DCE | 30 min | 76 | 5 |
| 10 | PTH4 | Toluene | 24 h | 3 | 81 (79)[d] |
| 11 | PTH3 | Toluene | 24 h | 4 | 76 |
| 12 | PTH2 | Toluene | 24 h | 82 | 11 |
| 13[e] | $fac$-Ir(ppy)$_3$ | Toluene | 24 h | 61 | 12 |
| 14 | - | Toluene | 30 min | 2 | 0 |
| 15 | - | Toluene | 24 h | 4 | 0 |
| 16[f] | PTH2 | Toluene | 30 min | 0 | 0 |
| 17[g] | PTH2 | Toluene | 30 min | 78 | 5 |
| 18[h] | PTH2 | Toluene | 24 h | 3 | 0 |
| 19[h] | - | Toluene | 24 h | 3 | 0 |

[a]Conditions: **1a** (0.4 mmol), **2a** (0.2 mmol), photo-catalyst (5 mol%), solvent (2 mL), degassed condition, irradiation with LEDs light ($\lambda_{max}$ = 390 nm) with 100% intensity at 30–35 °C; after completion of photo-reaction, 1 equivalent AlCl$_3$ (0.2 mmol) was added at 0 °C and stirred for 1 h; [b]AlCl$_3$ is necessary only facilitate $S$-cyclization; [c]Crude $^1$H NMR yield (%) using 1,1,2,2-tetrachloroethane as internal standard; [d]Isolated yield; [e]1 mol% catalyst loading, irradiation with LEDs light ($\lambda_{max}$ = 450 nm); [f]Reactions performed in dark; [g]Reactions performed in open air; [h]Reactions performed at 60 °C.

well as ribose-derived sugar substrate (**3ao**) in good yields (66%–78%), thus opening up the possibility of late-stage modification of biologically active molecules using this toolkit.

We further surveyed the applicability of this catalyst-controlled chemodivergent reaction by assessing the substrate scope of 2-thiopyrrolidinone synthesis using conditions B (Fig. 4)[45,46]. For aromatic alkenes, all the substrates reacted well, leading to the corresponding thiopyrrolidinone scaffolds (**4a-r**) in good to excellent yields (51%–79%), barring **4m,n** providing modest yields of 36% and 32% respectively possibly due to the deterioration of the alkene under the reaction conditions as mentioned above. Expectedly, unactivated aliphatic alkenes (1-dodecene, methylenecyclohexane) and acrylates proved to be unsuccessful, freezing at the carbothiocyanation stage (i.e., 1$^{st}$ cycle of ATRA only operative) due to the difficulty in photoredox-isomerization in 2$^{nd}$ cycle through oxidative radical-polar-crossover (*vide infra*)[44]. Guided by the above failure, we postulated that electron-rich aliphatic alkenes might facilitate the above isomerization through SET-oxidation, to provide the expected *N*-heterocycles. This was indeed the case with a range of electron-rich alkenes, such as vinyl ethers (**4s-u**) and vinyl amine (**4v**), leading to the desired thiopyrrolidone products in good yields (64%–78%). In the case of **4s**, a considerable amount of aldehyde **4s'** is formed possibly due to the generation of α-ethoxy carbocation during the oxidative polar crossover step[47]. Ene-carbamate (**4w**), and enamides (**4x-z**) were also suitable in the current method, although the yields were moderate (40%–63%) due to polymerization of such alkenes under the reaction conditions. Variations of malonate coupling partners were readily implementable to their desired products (**4aa–ad**) in good yields

(56%–73%). Moreover, the expedient transformation of the intricate olefins derived from estrone (**4ae**), vitamin E (**4af**), L-valine (**4ag**), and D-glucose (**4ah**) further showcased both the mild nature of this protocol along with the unique 'on-demand' selective diversification of medicinally interesting molecules. The formation of *N*-heterocycles was unambiguously proved by X-ray crystallography of **4k** and **4z**.

This chemoselective catalytic manifold involving TCMs was also extended to a range of other C-centered radicals (Fig. 5). A variety of thiocyanomethyl reagents (**2f-j**) substituted with carbonyl (-keto, -ester) or non-carbonyl (-nitrile, -benzyl) functionalities were effectively engaged with styrene as a bifunctional reagent, providing versatile carbo-thiocyanates (**5a-e**) and carbo-isothiocyanates (**6a-e**) with high synthetic value[3,4,6]. Further investigations on tertiary alkyl thiocyanate reagents allowed the incorporation of quaternary center γ- to thiocyanate (**5f-h**) and isothiocyanate products (**6f-h**) in synthetically useful yields (49%–84%). Moreover, Fluorinated alkyl-thiocyanate **2n** also successfully led to chemo-divergent products **5i** and **6i** with good yields (69%–78%). A few thiocyanates such as 2-thiocyanatopropane and (thiocyanatomethylene)dibenzene (See incompatible substrates list, Supplementary Table 6), however, could not be harnessed successfully into this manifold. While the former is an aliphatic thiocyanate having a reduction potential which is well beyond the scope of the catalysts in discussion, the later is easily oxidized to a highly stabilized benzhydryl carbocation and rather undergoes self-isomerization to isothiocyanate.

We were also curious to explore other heavier chalcogenonitriles, like hitherto unexplored ambient reactivity of selenonitrile which could possibly be tamed in a similar way by our strategy. We were

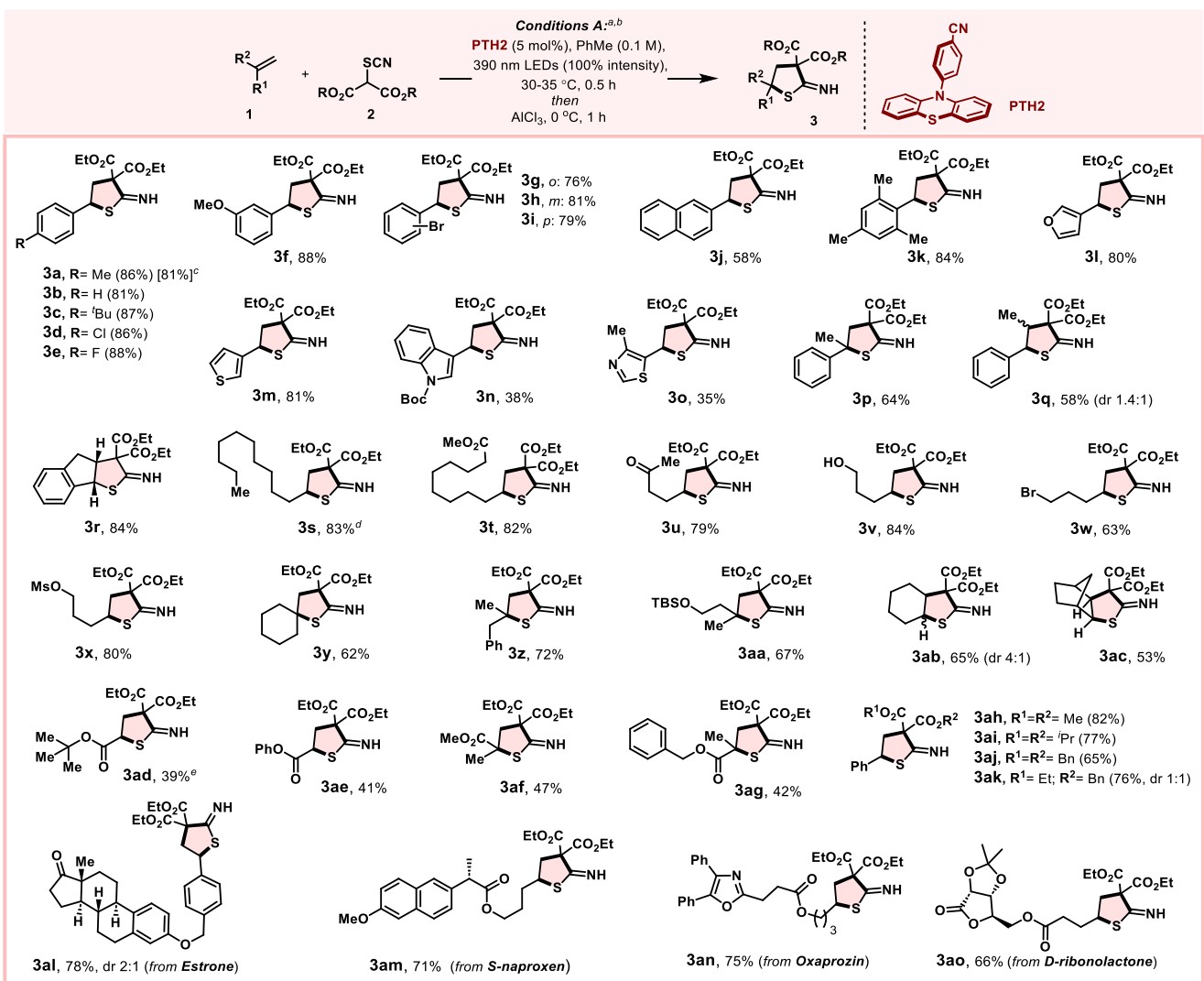

**Fig. 3 | Scope of the photochemical synthesis of 2-imino-tetrahydrothiophenes.** [a]Reaction conditions A: **1** (0.4 mmol), **2** (0.2 mmol), **PTH2** (5 mol%), and degassed toluene (2 mL) under argon with irradiation of LEDs ($\lambda_{max}$ = 390 nm) with 100% intensity at 30–35 °C for 30 mins. Then add AlCl$_3$ (1 equiv.) at 0 °C for 1 h. [b]isolated yield. [c]for gram scale reaction. [d]aliphatic alkene used (4 equiv.) for all cases, reaction time 1 h. [e]acrylate used (1 equiv.) for all cases, reaction time 1 h.

delighted to observe the selenocyanatomalonate **2o** undergo chemoselective photoredox annulation with various vinyl arenes leading to *Se*- (**7a-e**) and *N*- (**8a-e**) heterocycles with moderate to good yields (44%–72%) under conditions A and B, respectively (Fig. 5). Selenocyanatomalonate **2p** containing the bulky isopropyl ester groups was also used successfully as a bifunctional reagent in this process. Moreover, the photocatalytic switch was also effective for phenacyl selenocyanate **2q** to access γ-keto–selenocyanate (**7g**) and –isoselenocyanate (**8g**), classes of useful building blocks for the assembly of selenium-containing heterocycles and peptidomimetics[48,49]. Overall, these studies certified the generality and broad applicability of our current strategy in controlling ambient reactivity of chalcogenonitriles.

The synthesized *N*-heterocyclic scaffolds (both thiopyrrolidones **4** and selenopyrrolidones **8**) are normally stable at room temperature. However, their substituted hydrothiophene **3** and hydroselenophene **7** counterparts are not bench stable and require cooler storage temperatures.

The efficiency of this photo-annulation reaction could be further highlighted by its scalability, ease, and simplicity. The reaction between diethyl thiocyanomalonate **2a** and 4-methylstyrene **1a** was accomplished on a 6 mmol scale affording **3a** (1.63 g, 81%, Fig. 3) and **4a** (1.45 g, 72%, Fig. 4) under conditions A and B respectively,

demonstrating the robustness of the process. Interestingly, the reaction of 4-methylstyrene **1a** with diethyl bromomalonate and ammonium thiocyanate under slightly modified conditions also delivered the desired chemo-divergent products **3a** (63%) and **4a** (57%), albeit less yield compared to the corresponding bifunctional thiocyanate **2a**. This modular three-component recipe further enriches this developed method to access hydrothiophenes and pyrrolidine heterocycles from ready-stock materials by just regulating the photocatalyst (Fig. 6A). To demonstrate the utility of this method, we manipulated the 2-imino-hydrothiophene and 2-thiopyrrolidone products (Fig. 6B). For example, acid hydrolysis of **3b** resulted in thiolactone **9**, a useful monomeric skeleton for polymerization[50]. Treatment of geminal diester **3b** with catalytic Lewis acid leads to 2-amino-4,5-dihydrothiophenes **10** through a decarboalkoxylation-isomerization sequence. Notably, highly intricate spiro thia-oxazete[51] **11** and bridged this bicycle **12** were synthesized via inter- and intra-molecular nucleophilic addition-annulation at the imine center of **3 v** and **3aa**, separately. Oxidation of 2-thiopyrrolidone **4b** to the corresponding pyrrolidone **13** went cleanly with *m*CPBA. In addition, the alkylation of **4b** with dimethyl sulfate gave way to thioimidate **14** in good yield. The synthetic utility of free thiocyanates and isothiocyanates was also investigated. Nucleophilic substitution of diphenylphosphine oxide on thiocyanate

**Fig. 4 | Scope of the photochemical synthesis of 2-thiopyrrolidinone.** [a]Reaction conditions B: **1** (0.4 mmol), **2** (0.2 mmol), **PTH4** (5 mol%), and degassed toluene (2 mL) under argon with irradiation of LEDs ($\lambda_{max}$ = 390 nm) with 100% intensity at 30–35 °C for 24 h. [b]isolated yield. [c]for gram scale reaction.

compound **5f** produced phosphonothioate **15**, which is an essential component of therapeutic oligonucleotides[52]. The thiocyanate group of **5f** was also derivatized to trifluoromethyl thioethers **16** using TMSCF₃, which has been known to show high lipophilicity[53]. On the other hand, treatment of **6f** with benzamidine hydrochloride afforded 1,2,4-thiadiazoles **17** that can act as dual 5-lipoxygenase and cyclooxygenase inhibitors[54]. Finally, thiourea derivative **18** was prepared from the reaction of **6 f** with morpholine which has a wide range of pharmacological effects[55].

## Mechanistic studies

To gain mechanistic insights into the plausible mechanism, a series of control experiments were conducted (Fig. 7). Firstly, the radical nature of the reaction was indicated by inhibition of the reaction in the presence of TEMPO, with concomitant detection of TEMPO adducts **19**, **20**, and **21** (Fig. 7A). The same reactive intermediates (**19'** - **21'**) were also traced by reaction with 2-phenylimidazo[1,2-a]pyridine[56]. When α-cyclopropyl-4-chlorostyrene **22** was reacted with **2a** under standard conditions, ring-opened product **23** was isolated (Fig. 7B). Moreover, exposing diethyl 2,2-diallylmalonate **24** with **2k** to carbo-thiocyanation conditions resulted in the 5-exo-trig cyclized product **25**. Both these radical probe experiments confirm the initial formation of malonyl radicals in this process. To understand the source of the N-H proton in

**4a**, deuterated-TCM **2a-d** (73% D) was synthesized and reacted with **1a** under standard conditions with **PTH4**. Product **4a-d** was detected in crude ¹H NMR with 58% deuterium incorporation, indicating the methine hydrogen of **2a** is mainly supplying the N-H proton of product **4a** (Fig. 7C). The necessity of continuous photo-irradiation for the reaction was confirmed by a light-on/off experiment (Fig. 7D). UV/Vis measurements of individual reaction component and their combination do not hint towards the formation of an electron donor-acceptor complex between **1a** and **2k** (spectra I, Fig. 7E)[57]. In addition to the control results presented in Table 1 (entries 2, 14–19), the ineffective overlay between absorption spectra of starting thiocyanates (**2k** and **5f**) with emission spectra of LEDs ($\lambda_{max}$ = 390 nm) used, eliminates the possibility of C-S bond cleavage by direct light excitation of **2k** or **5f** (spectra II, Fig. 7E)[58,59]. In isothiocyanation event, control studies indicated that isomerization of **5f** to **6f** only took place in the presence of both catalyst and light (Fig. 7F). Moreover, exposure of benzyl thiocyanate **5f** (the first cycle product) to various nucleophiles (ethanol, 1,2,4-trimethoxybenzene) under conditions B afforded nucleophile-trapped **26a,b** along with the isomerized isothiocyanation product **6f** (Fig. 7G). This result indicated the intermediacy of a carbocation **5f'**, generated by an oxidative radical-polar crossover during the isomerization process with catalyst **PTH4** in the reaction. The non-isomerization of 1-Dodecene derived alkylthiocyanates **27** (Fig. 7G) is

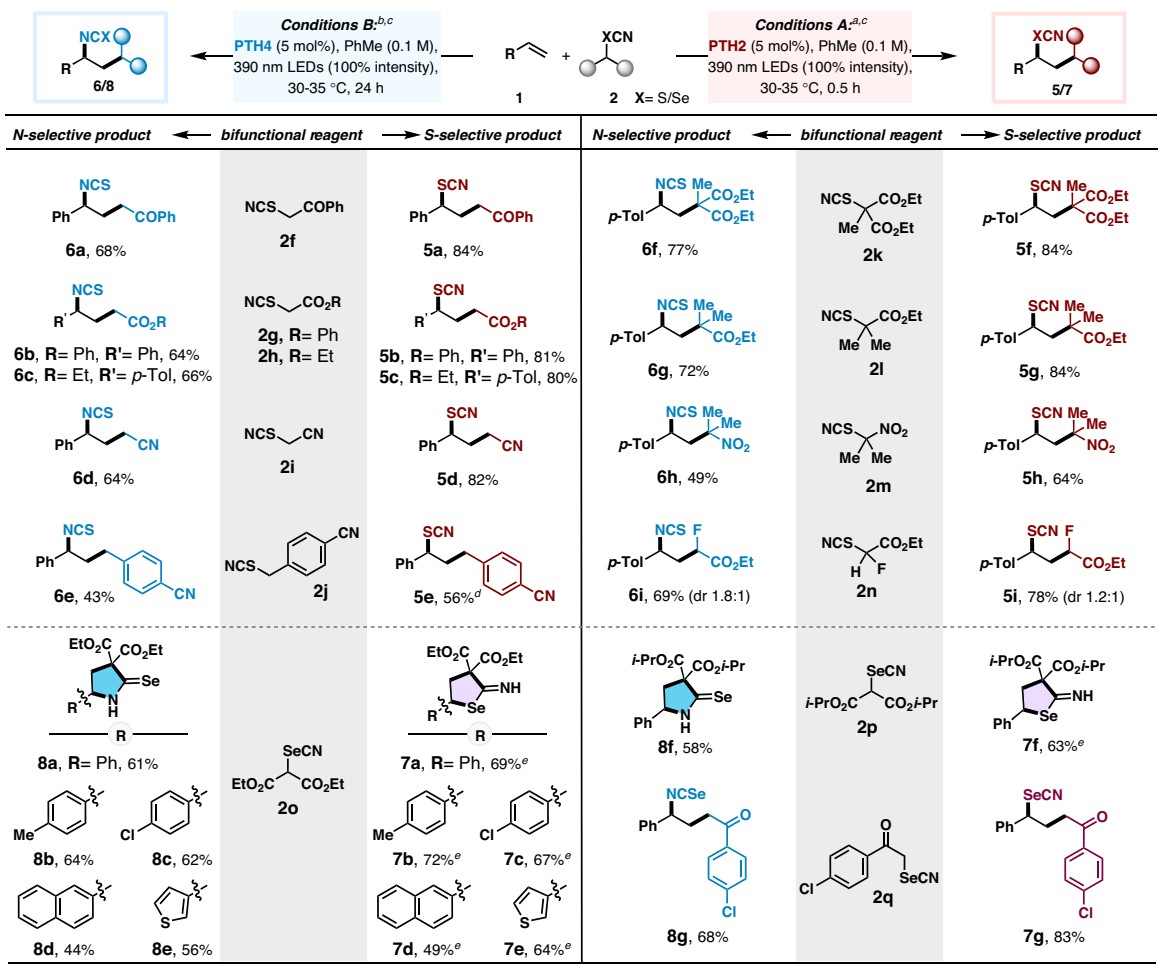

**Fig. 5 | Scope of the bifunctional chalcogenonitriles in photochemical chemodivergent reaction.** [a]Reaction conditions A: **1** (0.4 mmol), **2** (0.2 mmol), **PTH2** (5 mol%), and degassed toluene (2 mL) under argon with irradiation of LEDs ($\lambda_{max}$ = 390 nm) with 100% intensity at 30–35 °C for 30 mins. [b]Reaction conditions B: **1** (0.4 mmol), **2** (0.2 mmol), **PTH4** (5 mol%), and degassed toluene (2 mL) under argon with irradiation of LEDs ($\lambda_{max}$ = 390 nm) with 100% intensity at 30–35 °C for 24 h. [c]isolated yield. [d]*fac*-Ir(ppy)₃ (1 mol%) as catalyst. [e]after completion of photoredox reaction, AlCl₃ (1 equiv.) was added at 0 °C for 1 h.

tentatively attributed to their unfavorable reduction potential ($E^{red}$ = −2.39 V *vs.* SCE, see Supplementary Fig. 15) compared to **PTH4** ($E_{1/2}^{*}$ = −2.16 V *vs.* SCE), which cannot initiate the second photoredox cycle. To understand the trajectory of the cascade catalysis with **PTH4**, the reaction of **1a** with **2k** was monitored by ¹H NMR spectroscopy (Fig. 7H). In the first 20 minutes, complete conversion of the starting thiocyanomalonate **2k** took place, majorly to benzylthiocyanate **5f** (70%), along with a minor amount of isothiocyanate **6f** (22%) (Graph on first 1 h reaction, Fig. 7H). Subsequently, **5f** gradually converted to **6f** in 24 h time, suggesting that isomerization involving cascade cycle is the rate-limiting step (Graph on 24 h reaction, Fig. 7H). Stern-Volmer fluorescence quenching studies also indicated that both **2k** and **5f** are effective quenchers for excited state photocatalyst **PTH4** (Fig. 7I), while **2k** is the sole quencher for catalyst **PTH2** (Supplementary Fig. 26). To substantiate the potential mode of activation of the photocatalyst (via SET or EnT), the possibilities of both the modes were entertained by attempting each of the steps of our cascade reaction (thicyanation and isomerization) individually with a series of established energy transfer catalysts along with our initial SET capable catalysts (Fig. 7J). Computation of the triplet energies of the starting thiocyanates **2k** (44.9 kcal/mol) and **5f** (43.6 kcal/mol) revealed that all the photocatalysts listed in Fig. 7J (all having $E_T$ > 45 kcal/mol) rationally should provide both the desired products **5f** and **6f** (albeit to different extents), if an EnT pathway was at play at all. However as observed, the only productive catalysts were those whose reduction

potentials exceeded that of the starting thiocyanates **2k** ($E^{red}$ = −1.36 V vs SCE, entries 2, 5–8) and **5f** ($E^{red}$ = −1.83 V vs SCE, entries 5 and 8, see Supplementary Figs. 12 and 14 for electrochemical measurements), irrespective of their triplet energy. This bolstered our initial hypothesis of SET processes being responsible while eliminating any possibility of energy transfer.

From the detailed mechanistic studies and related literature reports[60–62], we proposed a mechanism to execute the formation of divergent products in Fig. 8A. Initially, a single-electron transfer from photoexcited catalyst PC* to thiocyanomalonate **2** creates thiocyanate anion and malonyl radical **A** which undergoes addition to the alkene **1** generating radical **B**. This proceeds to form **3'** via two possible pathways namely radical-radical cross-coupling (combination of **B** with thiocyanate radical, generated by oxidation of thiocyanate anion through SET oxidation during catalytic turnover) and radical chain transfer (kinetically feasible thiocyanate group transfer from thiocyanomalonate **2**). Exploration of these pathways by DFT calculations (Fig. 8B) revealed that the addition of malonyl radical **A** either onto 4-methylstyrene (aromatic conjugated olefin) or 1-dodecene (aliphatic variant) **1** (via **TS1**) proceeds irreversibly with a low energy barrier (8.8 and 12.3 kcal/mol, respectively) to deliver the intermediate **B**, downhill in energy by 16–21 kcal/mol. For the subsequent conversion of **B** to **3'** by selective radical-radical cross-coupling between **B** and SCN radical, though kinetically feasible based on the Ingold−Fischer "persistent radical effect"[63], would heavily compete with a significant number of

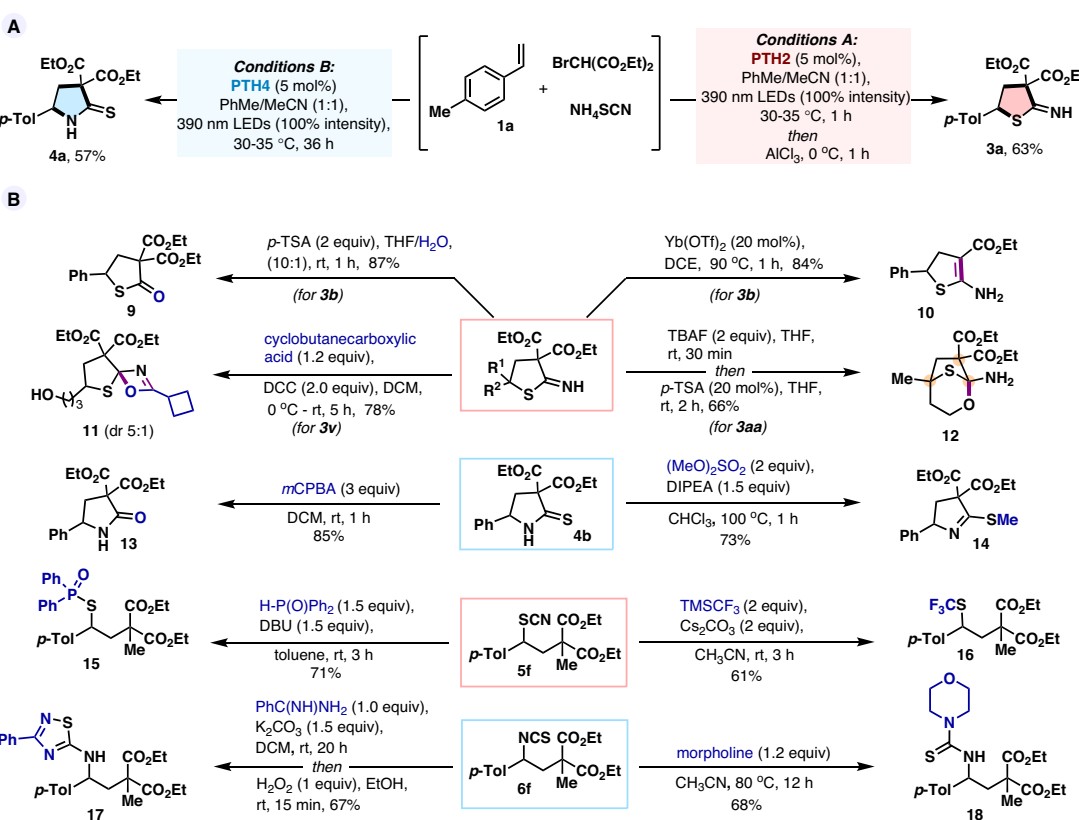

**Fig. 6 | Synthetic transformations. A** Three-component reaction. **B** Postsynthetic modifications.

highly probable unproductive pathways[64,65] thus diminishing its synthetic viability. A more plausible route to explain the high efficiency and selectivity observed would be envisioning intermediate **B** as a radical chain mediator, engaging with a second molecule of thiocyanatomalonate **2a** via **TS2** (with an energy barrier of 21.4 kcal/mol (R = p-Tol) and 15.7 kcal/mol (R = n-decyl). This leads to the metastable compound **3'** while the resulting second malonyl radical is easily sequestered by the excess olefin in the reaction. The metastable **3'** can subsequently undergo Lewis acid assisted cyclization to generate the S-heterocycles **3**. For styrene-derived alkenes, the possibility of an additional radical-polar crossover mechanism via a stable benzylic carbocation was also explored (see, additional reaction mechanism in SI). However, the high quantum yield (φ) of the reaction for both aromatic (36.9, R = p-Tol) as well as aliphatic (25.2, R = n-decyl) alkenes validates radical chain pathway being the major contributor[66].

When the reductively more potent **PTH4** is used, the previously formed thiocyanate **3'** undergoes a second SET event which splits it into radical **B** and thiocyanate anion. Hereon, if radical **B** is susceptible to being oxidized to a stable carbocation (as in case of styrenes, vinyl ethers, vinyl amines, etc.) by an oxidative polar-crossover event, the generated carbocation can be trapped by isothiocyanate anion to form the thermodynamically stable isothiocyanate **4'**, which subsequently cyclizes to afford the 2-thiopyrrolidone products **4**. DFT studies for the 2nd cycle suggest that the generation of thiocyanate anion and alkyl radical **B** from the photocatalytic reductive cleavage of alkyl thiocyanate **3'** takes place by a process that is exergonic by up to ~1.7 kcal/mol (R = p-Tol). This is followed by the endergonic polar crossover of radical **B** to carbocation **C** (by ~2.6 kcal/mol) which combines with isothiocyanate anion and undergoes spontaneous cyclization to N-Heterocycles **4** via intermediate **4'**. The minute amounts of **3'** produced by the competitive action of thiocyanate anion are also converted to **4** as it further takes part in the isomerization step.

In conclusion, we have successfully developed a photochemical chemodivergent route to incorporate thiocyanate and isothiocyanate functionalities onto olefins by controlling the ambient-SCN through catalyst modulation. The successful introduction of thio- and seleno-cyanomalonates as a bifunctional group transfer reagent in the photoredox process has enabled the redox-, and step-economic synthesis of 5-membered S-, Se- and N-heterocycles from feedstock alkenes through a cascade process. Due to the fundamental reactivity issues that are being solved here, this study on photocatalyst regulated tuning of ambient chalcogenonitriles will stimulate its wider application in radical chemistry research and as a synthetic tool in general. In addition, evidence from detailed control experiments along with density functional theory (DFT) calculations also provides a solid mechanistic backbone to the developed photo cascade strategy.

## Methods
### General procedure for photocatalytic 2-Imino-tetrahydrothiophenes synthesis
An oven-dried culture tube equipped with a magnetic stir bar was charged with **PTH2** (3 mg, 0.01 mmol, 5 mol%), thiocyanatomalonate **2** (0.2 mmol), and dry toluene (2 mL). The tube was sealed with a Teflon screw cap before olefin **1** (0.4 mmol aromatic olefin/ 0.8 mmol aliphatic olefin/ 0.2 mmol acrylate) was added to it. Then, the reaction mixture was degassed by Freeze-Pump-Thaw cycles with argon and irradiated at 30–35 °C with 390 nm LEDs (100% intensity) at a distance of ~5 cm for 30 min (1 h for aliphatic olefin and acrylate). A high-speed fan was used to maintain the temperature. After the completion of the ATRA reaction (confirmed by TLC), Aluminum chloride (27 mg, 0.2 mmol) was added to the ice-cold reaction mixture. After 1 h, 2 mL of ethyl acetate was added and quenched with saturated ammonium chloride solution (2 mL). The crude reaction mixture was extracted with ethyl acetate (2 × 2 mL), washed with brine (3 mL), and dried over anhydrous $Na_2SO_4$. The organic portion was concentrated, and the

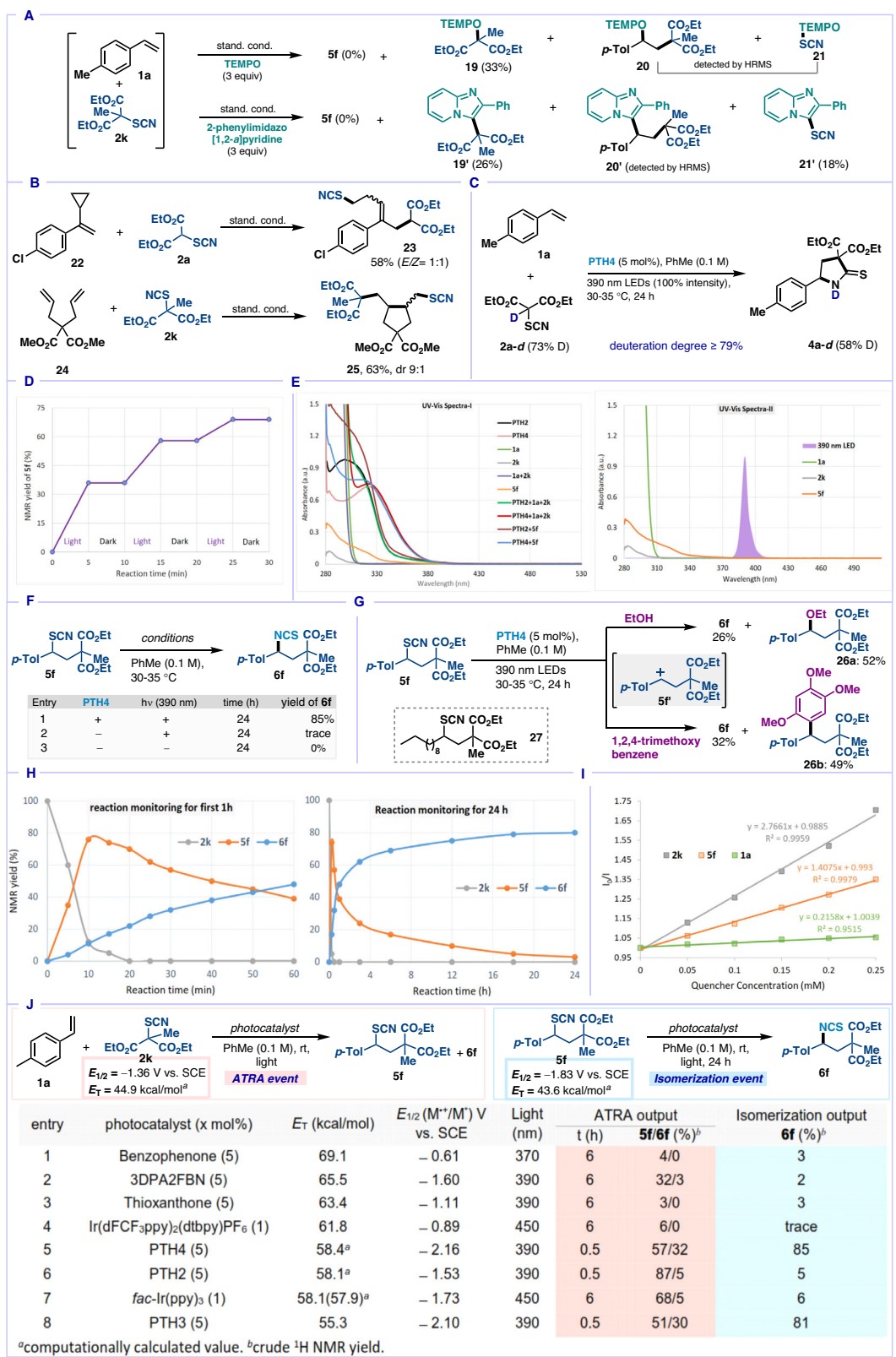

**Fig. 7 | Mechanistic studies. A** Radical trapping. **B** Radical probe. **C** Deuterium labeling experiment. **D** Light ON/OFF experiment. **E** UV/Vis absorption. **F** Control experiments on isomerization. **G** Isomerization intermediate evaluation. **H** Monitoring of cascade reaction (between **1a** and **2k**) by ¹H NMR. **I** Stern-Volmer quenching with **PTH4**. **J** Investigation into catalyst activation modes.

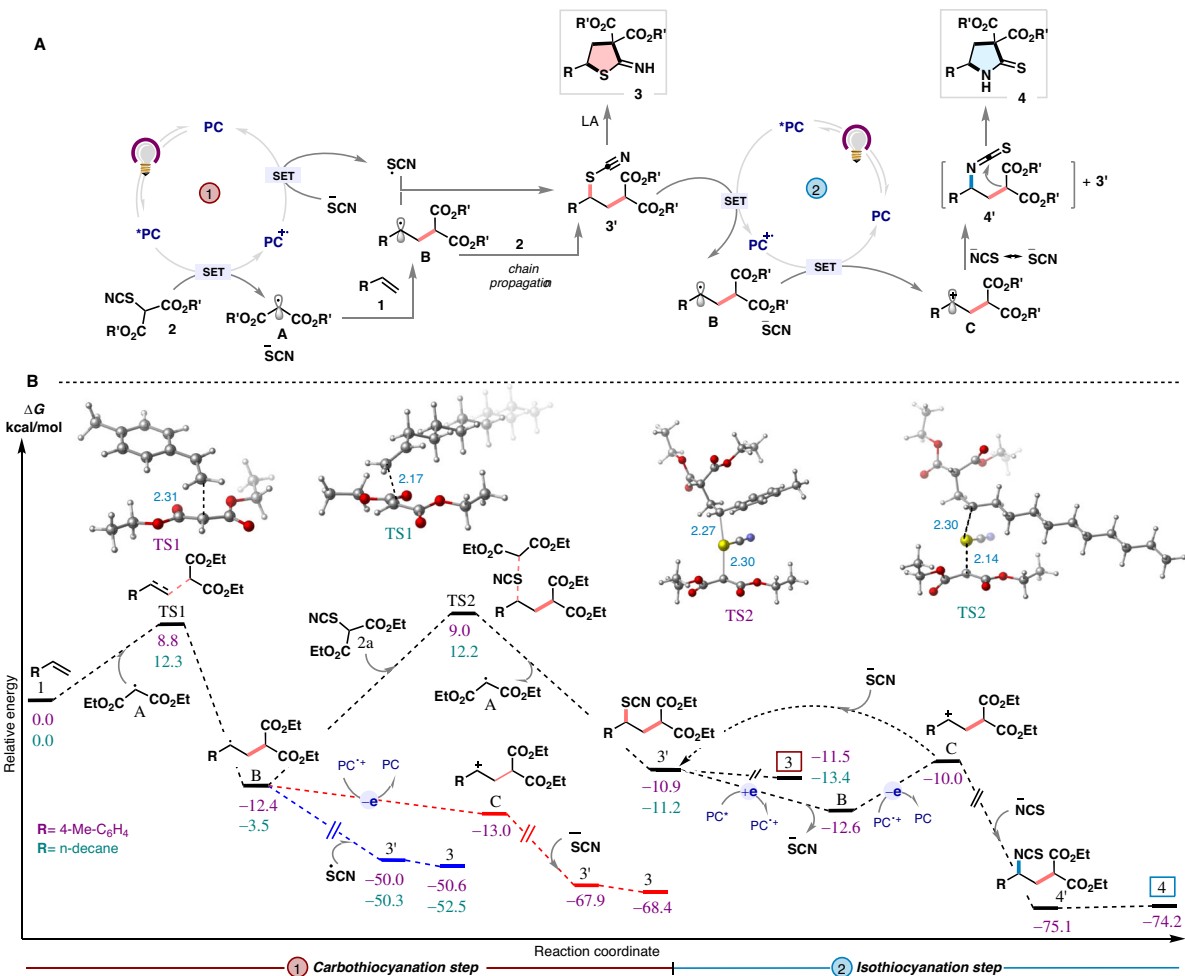

**Fig. 8 | Proposed mechanism supported by computational studies. A** Proposed mechanism. **B** Computational study. Free energy profile calculated at the [SMD(toluene) B3LYP-D3/def2-TZVPP] level of theory. Bond lengths are reported in Ångstroms.

residue was purified by silica gel column chromatography using EtOAc/petroleum ether as eluent to afford the corresponding 2-imino-tetrahydrothiophenes product **3**.

### General procedure for photocatalytic thiopyrrolidinones synthesis

An oven-dried culture tube equipped with a magnetic stir bar was charged with **PTH4** (3.7 mg, 0.01 mmol, 5 mol%), thiocyanatomalonate **2** (0.2 mmol), and dry toluene (2 mL). The tube was sealed with a Teflon screw cap before olefin **1** (0.4 mmol) was added to it. Then, the reaction mixture was degassed by Freeze-Pump-Thaw cycles with argon and irradiated at 30–35 °C with 390 nm LEDs (100% intensity) at a distance of approximately 5 cm for 24 h. A high-speed fan was used to maintain the temperature. After the completion of the reaction (confirmed by TLC), reaction crude was concentrated and purified by silica gel column chromatography using EtOAc/petroleum ether as eluent to afford the corresponding thiopyrrolidinones product **4**.

### Data availability

The data supporting the findings of this study are available within the paper and its Supplementary Information. The X-ray crystallographic coordinates for the structures reported (**4k** and **4z**) have been deposited at the Cambridge Crystallographic Data Center (CCDC), under deposition numbers CCDC 2164590 and 2182605, respectively. The data can be obtained free of charge from the Cambridge Crystallographic Data Center via www.ccdc.cam.ac.uk. Cartesian coordinates of computationally optimized geometries are available in

Supplementary Data 1. Further relevant data are available from the corresponding author upon request.

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

## Acknowledgements

Financial support from SERB (CRG/2021/004140), India is gratefully acknowledged. I.U.H. thanks DST-INSPIRE (IF160344) for a doctoral fellowship. R.L. thanks Prof. Pavel Hobza, Institute of Organic Chemistry and Biochemistry, Prague, Czech Republic, for providing the infrastructural facility for computational calculations. Prof. H. P. Nayek is acknowledged for helping with the X-ray crystallographic solutions. We thank Prof. S. K. Padhi for providing access to his instrumentation. We also thank Prof. Nan Zheng at the University of Arkansas for helpful discussions.

## Author contributions
S. M. conceived the concept, supervised the project, and drafted the manuscript. I. U. H. performed most of the experiments and mechanistic study with the help of A.S. S.P., and S. R. C. took part in the preparation of some of the starting materials. R.L. performed the computational studies. All authors discussed the results and contributed to the final paper.

## Competing interests
The authors declare no competing interests.
