## [Peer Review File · Nature Communications]

Photocascade Chemoselective Controlling of Ambident Thio(Seleno)cyanates with Alkenes via Catalyst ModulationReviewers' comments:

Reviewer #1 (Remarks to the Author):

In this manuscript, Maity and co-workers describes an efficient photoredox strategy for the controlled incorporation of thiocyanate and isothiocyanate functionalities onto olefins with thiocyanomalonates as a novel bifunctional group transfer reagent, which upon sequential cyclization enabled the atom-, redox-, and step-economic synthesis of hydrothiophene and pyrrolidine heterocycles. The authors have also expanded the scope of this chemoselective reaction, such as aliphatic alkenes, pharmaceutically relevant molecules, other thiocyanomethyl reagents and selenocyanatomalonate, to construct carbo-thiocyanates, carbo-isothiocyanates, N/S/Se-heterocycle frameworks which could be easily transformed into various value-added products. Furthermore, the authors have performed a series of mechanistic studies to support their proposed mechanism, including radical-trapping experiment, radical clock experiment, nucleophile-trapping experiment, determination of quantum yield, electrochemical measurements, ¹H-NMR spectroscopy, light on-off experiment, UV-Vis experiments, fluorescence quenching experiment, and relevant control studies. Given the importance of this work, I recommend publication in Nature Communications after major revisions. Specific comments and questions are provided below:

Paper:

(1) Page 2, Line 38: Please change “allowsdifferent” into “allows different”.

(2) Page 8, Scheme 3: Please provide diastereomeric ratio (dr) of compounds 13, 14, 5d, 6d, 3z-3ac, and 4x-4aa;

(3) In substrate scope section, could other heteroaryl-styrenes (e.g. pyridyl- and indolyl- styrenes) and electron-deficient alkenes be suitable for these two transformations?

(4) In mechanism section, when using these two catalysts (PTH2 and PTH4), is it possible that thiocyanate 2a could proceed via C-S bond homolysis (energy-transfer, EnT) to generate the corresponding radical for further transformation?

Supporting Information:

(1) Most of NMR spectra have minor unattributed impurities (1ae, 2a, 3c, 3i, 3r, 3w, 3y, 3z, 4s, 4w, 4x, 4aa, 5d, 7b, 7d, 7h, 8a, 9a, 9b, 18, 19, and so on). Please, address.

(2) Some of NMR data does not have sufficient signal/noise ratio, such as 3g, 3k, 3l, 3w 4e, 4aa, 5c, 9a, and so on. Please reacquire.

(3) Why are there two sets of peaks in ¹H-NMR of compounds 3g, 3k, and 5c?

Reviewer #3 (Remarks to the Author):

Maity and co-workers have described the photocascade chemoselective controlling of ambident Thio(Seleno)cyanates via electron transfer. Overall, the work has been nicely explored the reactivity of thiocyanate by generating the thiocyanate radical. The novelty of this protocol lies in reductive cleavage of C-S bond via single electron transfer and isomerization of thiocyanate to isothiocyanate under photocatalytic reaction conditions.

On the other hand, the addition of thiocyanate radical to alkene is well studied and the second acid catalysed cyclization step is also well known (Green Chem., 2015,17, 3515-3520; ACS Sustainable Chem. Eng. 2019, 7, 16, 14009–14015; Org. Biomol. Chem., 2019,17, 2232-2241).

Importantly, the pathways of C-S bond cleavage and isomerization of isothiocyanate is still unclear. The CV and Stern-Volmer experiment are not sufficient enough to provide the insight into the C-S bond cleavage and ambident reactivity of thiocyanate.

Based on these concerns and considering the lack significant novelty of the protocol, this reviewer found this manuscript not suitable for the publication in Nature communication.

1. Scheme present in the table-1 is not appropriate, the product presented in the table 1 reaction is the product of the second step and the condition of the second step is not presented in the reaction. The final product formations happen in two steps and it is not depicted in the reaction of table 1 second step reaction condition needed to be included.

2. Table-1, entry 14, indicating that the final product formation (2%) is just in 30 min irradiation of 390 nm light without photocatalyst. However, the author did not present the same reaction data after long reaction time such as after 24/48 h. This 2% yield of the final product in just 30 min suggests us to think critically about C-S bond cleavage of compound 2. Is this C-S bond cleavage because of the 390 nm light irradiation.? or is it the heat generated because of the 390 nm light irradiation inside the reaction responsible for such C-S bond cleavage.? Control experiments need to

be performed to address this product formation such as; same reaction for 24 h or 48 h without photocatalyst with 390 nm light and secondly the reaction at higher temperature 60 oC without irradiation of light with and without photocatalyst. (report based on C-S bond cleavage (α - to carbonyl) by simple irradiation of light - Am. Chem. Soc. 2017, 139, 40, 14315–14321; Chem. Commun., 2023, 59, 5343)

3. Is this C-S bond cleavage because of the single electron reduction.? or is it the energy transfer from the catalyst to molecule 2 triggering a homolytical bond cleavage resulting in two stable radical species? This aspect has not been considered, this ambiguity of energy transfer or electron transfer needs to be clarified. This reviewer suspects that one of the products formation could be because of the energy transfer because the thiocyanation and isothiocyanation are known to be kinetically and thermodynamically controlled.

4. This protocol is not explored for the electron deficient double bond except example 3y more such examples need to be explored.

5. NMR spectra of some compounds are not pure such as 3a' , 3c, 3i, 3w and 9a.

Comments

The manuscript entitled “Photocascade Chemoselective Controlling of Ambident Thio(Seleno)cyanates with Catalyst Selective Electron Transfer” by Maity and co-workers disclosed catalyst-regulated photoredox strategy for installing thiocyanates and isothiocyanates in a controlled chemoselective fashion by manipulating ambident-SCN through catalyst selective electron transfer. The key features are selectivity, atom-, and pot-economical and direct access to hydrothiophene and pyrrolidine heterocycles. The results presented herein build upon their previous work regarding visible light-promoted alkyl halides enabling photochemical alkylamination of olefins [Chem. Commun., 2022, 58, 8400]. Although synthesis of hydrothiophene and pyrrolidine derivatives from under modified reaction conditions by using diethyl 2-thiocyanatomalonate from alkenes will not improve novelty and impact of the reaction in the synthetic methodology. The extension approach of this manuscript is selective cyclization depends on photocatalyst compared to their previous work. I strongly believe, novelty of the reaction does not meet the high criteria of Nature Communications, it may be suitable in other specialized organic journals. In addition, the scope and the way the entire manuscript presented lacks the necessary thoroughness. However, the following points need to be considered before submitting them elsewhere.

Manuscript:

1. In the introduction, the authors stick mainly on the selectivity of thiocyanates and their reactivity. I think more detailed discussions are necessary by inserting schemes of the previous work on alkenes or other counter species under traditional or visible-light.
2. The references cited in the paper are very confusing. Some of them are not related to the content described in the paper, and some are not representative. It indicates that the authors did not strictly review the cited literature. The authors should check and confirm all the references in the paper, otherwise these errors will mislead the readers.
3. Did authors use any cooling system for their photoreactor? This information should be clearly mentioned in manuscript and supporting information. By using blue LED lamps, it easily provides higher temperature in longer reaction time. Authors should provide a clear

Reviewer #2 (Remarks to the Author):

- picture of their photoreaction setup in the supporting information with Blue LED details (other technical details of Blue LED). In addition, how many Watt lamps used in the reaction?
4. Would the reaction work with internal alkenes? Would the reaction be applicable to terminal/internal alkynes under the optimized reaction conditions?
 5. Does the LEDs light intensity have an impact on the yield of the product? What will happen if the reaction continues for a longer time with photocatalyst PTH2?
 6. The authors should test the following precursors to showcase the broad scope.in the with respective alkenes under the optimized reaction conditions.

7. Can the authors exclude the activity of an EDA in the reaction mechanism? Please see e.g.: J. Am. Chem. Soc. 2020, 142, 12, 5461–5476.
8. Please remove the arrow in the intermediate C because it is confusing with generation of intermediate B via SET process.
9. To justify only photocatalyst absorption of Blue LED absorption or exclude the EDA complex, authors should provide an absorption spectrum of all the starting materials or in combination.
10. In mechanistic studies, one reaction needs to be conducted by using 1,6-diene as a radical acceptor.
11. The authors proposed a reaction mechanism for the direct excitation of photocatalysts. The absorption spectra of the compound and the emission spectra of the light source should be reported in the main manuscript overexposed to demonstrate the possible direct excitation of the compound based on UV-vis spectra. And also move the light on-off experiments to the main manuscript.

12. Detailed mechanistic evidence is necessary to prove the reaction mechanism especially from intermediate **C** and **E** (I believe, this is a stable compound and isolable at low temperature or short reaction time).
13. To prove hypothesized reaction mechanism, authors should conduct DFT studies to define the key intermediates, Sulphur and nitrogen attack competing pathways.
14. The presentation of schemes is congested in the manuscript. Please redraw or rearrange for a better presentation.
15. There are many typo and grammatical errors in the manuscript. The authors should revise it carefully. English polish is also necessary.
16. Cite the following related papers on photoinduced reactions; a) ACS Org. Inorg. Au 2022, 2, 435–454; b) Nat. Commun., 2022, 13, 2345.

Supporting Information:

17. Some spectra feature very broad signals (**3g**, **3k**, **3i** and etc) in ^1H NMR. Please find this unusual outcome compared to other products in the manuscript.
18. In the manuscript, the authors showed mixture of diastereomeric isomers (dr) in compounds **4n**. However, the ^1H and ^{13}C NMR showed only single isomer. Please check it.
19. Few spectra are blurring (example; **3z**, provide high resolution spectra. In addition, some of the final products are not pure (for example: **5ad**, **5ac** and **5ab**). Repurify and recalculate the yields.
20. Some ^1H NMR and ^{13}C NMR spectra clearly show impurities (example **1ae**, **3y** and **7b** etc). Re-purification is required.
21. The ^{13}C spectra indication should be decimal, f.i. 177.6 instead of 177.57.
22. In Supporting Information, the author should include the solvent and NMR frequencies (MHz) in the copies of ^1H , ^{13}C -NMR spectra.
23. In compound **5d** spectra (^1H and ^{13}C NMR), why authors give integration wrongly because there is a clear splitting in the peaks (for example at 3.0 ppm and 1.25 ppm). I believe this is because of the mixture of regio products (Sulphur and nitrogen attacked products). And also, the ^{13}C NMR carbon count is too much. Please check it carefully in all the compounds.

Response:

Reviewer #1

In this manuscript, Maity and co-workers describes an efficient photoredox strategy for the controlled incorporation of thiocyanate and isothiocyanate functionalities onto olefins with thiocyanomalonates as a novel bifunctional group transfer reagent, which upon sequential cyclization enabled the atom-, redox-, and step-economic synthesis of hydrothiophene and pyrrolidine heterocycles. The authors have also expanded the scope of this chemoselective reaction, such as aliphatic alkenes, pharmaceutically relevant molecules, other thiocyanomethyl reagents and selenocyanatomalonate, to construct carbo-thiocyanates, carbo-isothiocyanates, N/S/Se-heterocycle frameworks which could be easily transformed into various value-added products. Furthermore, the authors have performed a series of mechanistic studies to support their proposed mechanism, including radical-trapping experiment, radical clock experiment, nucleophile-trapping experiment, determination of quantum yield, electrochemical measurements, ¹H-NMR spectroscopy, light on-off experiment, UV-Vis experiments, fluorescence quenching experiment, and relevant control studies. Given the importance of this work, I recommend publication in Nature Communications after major revisions.

>> We thank the Reviewer for the encouragement and appreciate his enthusiasm for the publication.

Specific comments and questions are provided below:

Paper:

(1) Page 2, Line 38: Please change “allowsdifferent” into “allows different”.

>> The necessary corrections have been made.

(2) Page 8, Scheme 3: Please provide diastereomeric ratio (dr) of compounds **13**, **14**, **5d**, **6d**, **3z-3ac**, and **4x-4aa**;

>> Thank you very much for the valuable comment. We have added the diastereomeric ratio (dr) of the following compounds: **13** (**11**), **5d** (**3ah**), **6d** (**4ac**), and **3z** (**3ai**). The rest of the compounds appeared as single isomeric product in ¹H NMR of crude reaction mixture and accordingly not mentioned. [in bracket, new numbering].

(3) In substrate scope section, could other heteroaryl-styrenes (e.g. pyridyl- and indolyl- styrenes) and electron-deficient alkenes be suitable for these two transformations?

>> We are thankful for the suggestion. Apart from the existing furan (**3i**) and thiophene (**3m**, **4l**) heterocycles, we have conducted the reaction with few other heteroaryl-styrenes and electron-deficient alkenes with our developed methods and obtained the desired products. In the case of 2-vinyl pyridine, we have not isolated any desired outcome. We have summarized the isolated products with yields for the mentioned alkenes.

alkenes	Reaction with PTH2	Reaction with PTH4
---------	--------------------	--------------------

					A complex mixture was observed after the reaction with malonate 2a in standard conditions, which is difficult to isolate.	A similar type of complex mixture was observed.
		Not applicable. As illustrated in Scheme 7, isomerization of –SCN to –NCS in 2 nd catalytic cycle involves carbocation intermediate, generated by oxidation of carbon-centered radical through radical-polar-crossover mechanism. But alkyl radical oxidation adjacent to the ester (for α -thiocyanato ester) is relatively difficult (See: X.-J. Tang, W. R. Dolbier, Jr., Angew. Chem. Int. Ed. 2015 , 54 , 4246-4249.) and reaction with acrylates lead to thiocyanate/S-cyclization products.

(4) In mechanism section, when using these two catalysts (**PTH2** and **PTH4**), is it possible that thiocyanate **2a** could proceed via C–S bond homolysis (energy-transfer, EnT) to generate the corresponding radical for further transformation?

>> Thank you very much for asking us to explore this possible facet of the mechanism. To get better insight about the energy-transfer mode of catalyst activation for C–S bond cleavage, we have computationally calculated the triplet energy (E_T) of both substrates **2k**, **5f** and catalysts **PTH2**, **PTH4**, *fac*-**Ir(ppy)₃**. Additionally, the electrochemical measurements ($E_{1/2}$) of corresponding substrates and catalyst were also done. Then the possibility of both EnT and SET were examined extensively by carrying out each of the steps of the cascade reaction individually with a series of catalysts that have been traditionally known for energy transfer, along with our initial SET capable catalysts. Even though all the catalysts in the series had triplet energies higher (>45 Kcal/mol) than the starting thiocyanates **2k** (44.9 kcal/mol) and **5f** (43.6 kcal/mol) thus making them capable of driving the reaction by C–S

bond homolysis, only those catalysts with reduction potential higher than the substrates **2k** ($E^{\text{red}} = -1.36$ V vs SCE, entries 2, 5-8) and **5f** ($E^{\text{red}} = -1.83$ V vs SCE, see SI for Electrochemical Measurements), were productive (entries 5 and 8). This proved that triplet energies and by inference the EnT route had no role to play in this cascade, with SET being the plausible pathway.

Investigation into catalyst activation modes

entry	photocatalyst (x mol %)	E_T (kcal/mol)	$E_{1/2} (M^+/M^-)$ V Vs. SCE	Light (nm)	ATRA output		Isomerization output
					t (h)	5f/6f (%) ^b	6f (%) ^b
1	Benzophenone (5)	69.1	-0.61	370	6	4/0	3
2	3DPA2FBN (5)	65.5	-1.60	390	6	32/3	2
3	Thioxanthone (5)	63.4	-1.11	390	6	3/0	3
4	Ir(dFCF ₃ ppy) ₂ (dtbpy)PF ₆ (1)	61.8	-0.89	450	6	6/0	trace
5	PTH4 (5)	58.4 ^a	-2.16	390	0.5	57/32	85
6	PTH2 (5)	58.1 ^a	-1.53	390	0.5	87/5	5
7	fac-Ir(ppy) ₃ (1)	58.1(57.9) ^a	-1.73	450	6	68/5	6
8	PTH3 (5)	55.3	-2.10	390	0.5	51/30	81

^a computationally calculated value. ^b crude ¹H NMR yield.

The discussion has been added in the revised manuscript (mechanistic studies) and the table in Scheme 6J.

Supporting Information:

(1) Most of NMR spectra have minor unattributed impurities (**1ae**, **2a**, **3c**, **3i**, **3r**, **3w**, **3y**, **3z**, **4s**, **4w**, **4x**, **4aa**, **5d**, **7b**, **7d**, **7h**, **8a**, **9a**, **9b**, **18**, **19**, and so on). Please, address.

>> We are thankful for the comment. After carefully observing all NMR spectra, we have replaced the ¹H, ¹³C, and ¹⁹F NMR spectra with minor unattributed impurities. We have changed the spectra of compounds **1ae** (**1aj**), **2a** (**2a**), **3c** (**3c**), **3i** (**3i**), **3r** (**3t**), **3w** (**3y**), **3y** (**3ac**), **3z** (**3ai**), **4s** (**4u**), **4w** (**4y**), **4x** (**4ad**), **5d** (**3ah**), **7b** (**5b**), **7d** (**5e**), **8a** (**6a**), **9a** (**7a**), **9b** (**7b**), **18** (**16**), and **19** (**17**). In addition to these, we have also reacquired the spectra for compounds **3a'** (**3a'**), **3g** (**3g**), **3k** (**3k**), **3l** (**3l**), **4e** (**4e**), **5c** (**3ag**). Compound **7h** contains some persistent impurities which we have failed to remove to the best of our efforts. Hence, we have replaced it with another fluorinated compound **5i** prepared from the bifunctional thiocyanate **2n** and provided its pure spectra [in bracket, new numbering].

(2) Some of NMR data does not have sufficient signal/noise ratio, such as **3g**, **3k**, **3l**, **3w**, **4e**, **4aa**, **5c**, **9a**, and so on. Please reacquire.

>> We are thankful for the observation. After carefully observing all NMR spectra, we have replaced the ¹H, ¹³C, and ¹⁹F NMR spectra having insufficient signal/noise ratio. We have changed the spectra of compounds **3g** (**3g**), **3k** (**3k**), **3l** (**3l**), **3w** (**3y**), **4e** (**4e**), **5c** (**3ag**), **9a** (**7a**). [in bracket, new numbering]. In addition to these, all the reacquired spectra as mentioned in the reply to the previous comment have been obtained at better signal/noise ratio wherever necessary.

(3) Why are there two sets of peaks in ¹H-NMR of compounds **3g**, **3k**, and **5c**?

>> We are thankful for the observation. However, we have no idea what has caused this uncharacteristic splitting of the peaks in ¹H NMR of the above mentioned compounds. This most probably may be attributed to an instrumental error. We have reacquired the spectra of the mentioned compounds [**3g**, **3k**, and **5c (3ag)**] which now do not contain the above-mentioned issue. [in bracket, new numbering].

Reviewer #2

The manuscript entitled “Photocascade Chemoselective Controlling of Ambident Thio(Seleno)cyanates with Catalyst Selective Electron Transfer” by Maity and co-workers disclosed catalyst-regulated photoredox strategy for installing thiocyanates and isothiocyanates in a controlled chemoselective fashion by manipulating ambident-SCN through catalyst selective electron transfer. The key features are selectivity, atom-, and pot-economical and direct access to hydrothiophene and pyrrolidine heterocycles. The results presented herein build upon their previous work regarding visible light-promoted alkyl halides enabling photochemical alkylamination of olefins [Chem. Commun., 2022, 58, 8400]. Although synthesis of hydrothiophene and pyrrolidine derivatives from under modified reaction conditions by using diethyl 2-thiocyanatomalonate from alkenes will not improve novelty and impact of the reaction in the synthetic methodology. The extension approach of this manuscript is selective cyclization depends on photocatalyst compared to their previous work. I strongly believe, novelty of the reaction does not meet the high criteria of Nature Communications, it may be suitable in other specialized organic journals. In addition, the scope and the way the entire manuscript presented lacks the necessary thoroughness. However, the following points need to be considered before submitting them elsewhere.

>> We thank the reviewer for his efforts to review our manuscript, but at the same time, we are disappointed with the assessment of the manuscript by Reviewer 2.

Regarding the comment: “The results presented herein build upon their previous work regarding visible light-promoted alkyl halides enabling photochemical alkylamination of olefins [Chem. Commun., 2022, 58, 8400]”

>> *In our opinion, this is a wrong conclusion!* In this context, we have compared the current scheme with our previous work (Scheme I & II below). Our previous work involved “3-component oxidative radical-polar difunctionalization of alkenes”, while our current work is on “catalyst controlled 2-components chemodivergent difunctionalization of alkenes with a novel bifunctional reagent (as referred by Reviewer #1) through cascade catalysis.” One can’t compare any reaction associated with alkenes, involving difunctionalization, without considering the importance, innovations, and the concept involved in designing it. Firstly, identifying novel bi-functional reagents for atom-economical synthesis is an evolving field of research in recent times (Recent reports on bi-functional reagents: *J. Am. Chem. Soc.* **2023**, DOI: 10.1021/jacs.3c08512; *J. Am. Chem. Soc.* **2023**, *145*, 2364-2374; *J. Am. Chem. Soc.* **2022**, *144*, 15871-15878; *Nat. Rev. Chem.* **2021**, *5*, 301-321; , *J. Am. Chem. Soc.* **2021**, *143*, 2812-2821; *Nat. Commun.* **2019**, *10* (1), 4117; *Science* **2018**, *361*, 1369-1373, etc.). Herein, we introduced alkylthio(seleno)cyanates as a new bi-functional reagent for the first time. Secondly, controlling the ambident nucleophilicity of thiocyanates by photo-catalyst under mild conditions is another innovation that allows us to access two different series of heterocycles (*S*- and *N*-) from the same bi-functional reagent with complete control (Scheme II below). The design and success of this reaction depend on various key parameters (1→4, Scheme III below): understanding the inherent ambident reactivity profile of -S(Se)CN and translation of these physical parameters into favorable reaction parameters. We smartly harnessed the single electron transfer (SET) capability of photoredox catalysis to generate radical and ionic intermediate by cascade catalysis. By matching the kinetically fast photoredox radical-chain process, we synthesized carbothiocyanate products (adding ‘S’-side of -SCN), whereas by the thermodynamically slow radical-polar process, we accessed carbo-isothiocyanates (add ‘N’-side of -SCN). The innovation of new catalysts (PTH2, known but unexplored, and PTH4, our new synthetic

catalyst) controlled the selectivity between radical-chain and radical-polar process, ensuring chemo-selective products. A predicted catalyst switch is vital for ‘on-demand’ access to chemodivergent products. Here we creatively made a controlled catalytic switch by matching the redox profile of the catalyst with substrates, leading to chemoselective outcomes. Moreover, to the best of our knowledge, the ambident nucleophilicity of selenocyanates has never been studied before which provides the *Se*- and *N*-Heterocycles here. Our cascade catalysis approach in controlling ambident-SCN featuring a selective combination of sequential radical-chain and radical-polar processes by a single photocatalyst also has no precedent in literature. This work is an atom-, step-, and redox- economic reaction where ambident nucleophilicity of thio(seleno)cyanates has been precisely controlled by tuning it with the redox potential of the photocatalyst. Hence, the present work is conceptually completely novel and significantly advanced, which can’t be compared with our previous work or any other difunctionalization work.

I. Photochemical alkylation of olefins (*our previous work*)

II. 2-Component Group Transfer thiocyanation and isothiocyanation from alkylthiocyanate bi-functional reagent (*Unknown, this work*)

III. Reaction design, Catalyst innovation & Predicted outcome with catalyst switch: (*this work*)

Regarding the comment: “Although synthesis of hydrothiophene and pyrrolidine derivatives from under modified reaction conditions by using diethyl 2-thiocyanatomalonate from alkenes will not improve novelty and impact of the reaction in the synthetic methodology.”

>> *This is another misconception!* First of all, the reagent diethyl 2-thiocyanatomalonate is neither a commercial reagent nor the radical reactivity of this compound has ever been studied. Thiocyanomalonates have

previously been known to react under basic conditions to show some classical ionic reactivity (*J. Am. Chem. Soc.* **1939**, *61*, 1830; *Org. Biomol. Chem.* **2008**, *6*, 2995) at nucleophilic malonate and electrophilic nitrile centre of –SCN (centres 1 & 3 as depicted in Scheme IV below), but the radical reactivity of this class of compounds remains unexplored. We envisioned that this reagent might be intrinsically utilized as a group-transfer precursor bearing the malonate and the thiocyanate (center-1 and 2, respectively), which upon single electron transfer (SET), can generate the electrophilic malonyl radical amenable for addition to C-C double bonds with simultaneous formation of the nucleophilic thiocyanate for redox-neutral difunctionalization of alkenes. However, there are several challenges to overcome: i) the difunctionalization event should undergo fruitful ATRA in a regio-selective fashion, as alkylthiocyanate has never been explored as a group-transfer-agent; ii) Chemo-selectivity is another important aspect for reactions with thiocyanate group since thiocyanate anion shows ambident reactivity; iii) the cyclization should end up with a productive 5-membered ring, as nucleophilic removal of thiocyanate leads to a cyclopropyl ring.

IV. Thiocyanomaltonates reactivity modes:

V. 5-membered heterocycle synthesis: ionic vs radical approach

Moreover, 5-membered heterocycles synthesis is an attractive topic in current organic synthesis and is preceded by a series of fantastic literature reports, including *Acc. Chem. Res.* **2021**, *54*, 1528; *Angew. Chem. Int. Ed.* **2020**, *59*, 3385; *Angew. Chem. Int. Ed.* **2021**, *60*, 25825; *Angew. Chem. Int. Ed.* **2023**, *62*, e202214390. However, their synthesis is mainly limited to Lewis-acid catalyzed ionic [3+2] cycloaddition using donor-acceptor cyclopropanes (DACs), which impeded their broad applications, particularly for late-stage functionalization of complex molecules (Scheme V above). Our interest in synthesizing diverse heterocycles using bimolecular radical [2+2+1] heteroannulation from feedstock alkenes (aromatic, aliphatic, or complex alkenes) and Thio(seleno)cyanomaltonates provides the potential solutions to overcome the limitations of existing ionic methods. The robustness of the current radical method is demonstrated by the synthesis of 73 diverse functionalized heterocyclic (*S*-, *N*-, *Se*-) skeletons, including 8 complex molecules, which are not accessible easily with any other previously published works. The synthetic heterocycles have many utilities for making interesting molecules including 2-amino-(hydro)thiophene based drugs and functionalized heterocycles as delineated in scheme 5 in the manuscript.

Regarding “The extension approach of this manuscript is selective cyclization depends on photocatalyst compared to their previous work.”

>> *This is another wrong comment!* The present work is not a photocatalyst selective cyclization; rather, the work delineated here is photocatalyst selective chemodivergent -SCN/-NCS-installation through cascade catalysis. For *S*-heterocycles, the cyclization needs a Lewis acid treatment, whereas cyclization to *N*-heterocycle takes place

automatically under photocatalytic conditions through a domino difunctionalization-isomerization-cyclization sequence.

Overall, it is really disheartening that Reviewer #2 wrongly compares this draft with our previous work, and unfortunately undermines the novelty and impact of the current work in synthetic methodology development.

Manuscript:

1. In the introduction, the authors stick mainly on the selectivity of thiocyanates and their reactivity. I think more detailed discussions are necessary by inserting schemes of the previous work on alkenes or other counter species under traditional or visible-light.

>> We are thankful for your comment. As for your suggestion, we have made the necessary changes which have been included in the revised manuscript (Scheme 1C).

2. The references cited in the paper are very confusing. Some of them are not related to the content described in the paper, and some are not representative. It indicates that the authors did not strictly review the cited literature. The authors should check and confirm all the references in the paper, otherwise these errors will mislead the readers.

>> After a careful review of the citations and the references, we have replaced some of them with more impactful ones, and included some new references as well, the changes being highlighted in the revised manuscript. At the same time, we are also of the opinion that the previous references were definitely not out-of-place and were not included incorrectly as extensive search was carried out before insertion of each one of them.

3. Did authors use any cooling system for their photoreactor? This information should be clearly mentioned in manuscript and supporting information. By using blue LED lamps, it easily provides higher temperature in longer reaction time. Authors should provide a clear picture of their photoreaction setup in the supporting information with Blue LED details (other technical details of Blue LED). In addition, how many Watt lamps used in the reaction?

>> The details regarding the cooling system have already been mentioned multiple times in the the manuscript as well as supporting information (SI) file. To make matters clearer, we had already included a photograph of the reaction setup ‘including the high-speed fan’ in the original SI. We are surprised that the reviewer missed even a picture! Secondly, in multiple instances (point 3, point 9) the reviewer asks for details of blue LED. This work used mainly 390 nm, which is violet light, not blue light! Moreover, the information of light (Kessil LEDs, PR160L-390 nm, max 52W: page S3, SI) and emission spectra of light (Figure S19, Figure S22 in SI), are already there in SI. However, for any further technical information regarding such LEDs, kindly visit https://www.kessil.com/products/science_PR160L.php. During the course of this review process we performed some screening experiments newly, which involved blue LEDs. The mentioned LEDs are PAR38 12W blue LED providing 450 nm wavelength of light. This information has also been included in the revised SI (page S3).

4. Would the reaction work with internal alkenes? Would the reaction be applicable to terminal/internal alkynes under the optimized reaction conditions?

>> Again, this may have been erringly overlooked by the reviewer as we had already included three representative internal alkenes like 1,2-dihydronaphthalene, indene, norbornene and their desired products, [**3o** (**3q**) and **3x** (**3z**) in Scheme 3; **4n** (**4p**) and **4o** (**4q**) in Scheme 3 (Scheme 4)], along with yield in the original manuscript [in bracket, new numbering]. However, we have now segregated each type of product into its own scheme, making it easier for potential readers to follow. This reaction isn't applicable to terminal/internal alkynes under the optimized reaction conditions. After the reaction with phenylacetylene and diphenylacetylene, a complex mixture was observed which was unidentified. We have included these in the incompatible substrate list [Table S6 in SI].

5. Does the LEDs light intensity have an impact on the yield of the product? What will happen if the reaction continues for a longer time with photocatalyst PTH2?

>> We are thankful for the question regarding intensity. Yes, light intensity has an impact on the yield of the product. The progress of the reaction depends on the flux of the light. To address this, we have carried out the reaction at four ascending intensities, 25%, 50%, 75% and 100%. An upward movement of yield was observed along with the increasing intensity as follows. The corresponding table has been included in the Revised SI (Page 76 and Table S2).

Entry	Light intensity (%)	¹ H NMR yield of 5f/6f (%)
1	25	17/trace
2	50	46/trace
3	75	68/3
4	100	87/5

The long reaction times have already been documented in the optimization table (24 h vs 30 mins) with PTH2 were provides in Table 1, entries #12 vs #2).

6. The authors should test the following precursors to showcase the broad scope in the with respective alkenes under the optimized reaction conditions.

>> We are thankful for the suggestion. We have conducted the reactions of suggested alkyl-thiocyanates according to our developed methods. We have summarized the substrate-to-substrate comments and revised the manuscript accordingly.

Alkyl thio(telluro)cyanates	Output	Comments
2h	5c (80%) with PTH2 6c (66%) with PTH4	This alkyl-thiocyanate is a suitable substrate for our developed protocol. With 4-methylstyrene, it afforded the product 5c with 80% yield in the presence of photocatalyst PTH2 and 6c with 66% yield in the presence of photocatalyst PTH4. This result was incorporated in Table 2.
	NA	According to literature reports, the only available method for synthesizing the said tellurocyanate is by making use of KCN (Ref: Al-Masoudi, W. A. Sci. J. Univ. Zakho 1, 791-797 (2013)). The aforementioned synthesis could not be attempted due to KCN being placed under contraband category in India which has made its procurement from national or international vendors a very difficult task for us. So the scope of Tellurocyanide is out of our current study.

207810 ▶ Sigma-Aldrich

Potassium cyanide

★★★★★ (0) Write a review

ACS reagent, ≥96.0%

Linear Formula:
KCN

CAS Number: 151-50-8 Molecular Weight: 65.12 Beilstein: 3593645

EC Number: 205-792-3 MDL number: MFCD00011397 PubChem Substance ID: 329752024

NACRES: NA.21

Copy Link Email

All Photos (3)

Documents

SDS

COO/COA

Specification Sheet

❗ Product 207810 is not currently sold in your country. Contact Technical Service

 2s	 30 (57%)	Reaction was attempted with styrene and PTH4, however, only self-isomerization took place and we isolated the product 30 in 57% yield. Such benzhydryl radical can easily be oxidized to highly stabilized carbocations which was trapped by the isothiocyanate anion and leading the product 30.
 Complex mixture	NA	During its preparation, the desired thiocyanate is isomerized to its enol form and not isolable in its original form. [please see; (a). Liang, S. et al. Redox Active Sodium Iodide/Recyclable Heterogeneous Solid Acid: An Efficient Dual Catalytic System for Electrochemically Oxidative α-C–H Thiocyanation and Sulfenylation of Ketones. Adv. Synth. Catal. 360, 1444-1452 (2018); (b) Prakash, O. et al. α-Thiocyanation of carbonyl and β-dicarbonyl compounds using (dichloriodo) benzene–lead (II) thiocyanate. J. Org. Chem. 66, 2019-2023 (2001)].
 2q	NA	A complex, unisolable mixture was observed after reaction with styrene in standard conditions.
	NA	The reviewer mistakenly drew the same structure as above.
 2r	 29 (78%)	Malonate 29 (78%) was afforded instead of the desired product after the reaction with 4-methylstyrene under standard conditions. After photo-irradiation, destabilized diethyl 2-phenylmalonate radical/ doubly destabilized carbocation was formed, and quenching by the solvent afforded the mentioned product.
 2t	No reaction	No reaction occurred with styrene, due to the higher reduction potential of 2t ($E_{1/2} = -2.48$ V vs. SCE) than the existing photocatalysts (PTH2, $E_{1/2}^* = -1.53$ V and PTH4, $E_{1/2}^* = -2.16$ V vs. SCE).

7. Can the authors exclude the activity of an EDA in the reaction mechanism? Please see e.g.: J. Am. Chem. Soc. 2020, 142, 12, 5461–5476.

>> Thank you for the valuable comment. To investigate a detailed reaction mechanism, we have measured the UV-visible absorption spectra of the individual reaction components, and combined components with or without photocatalyst. However, there is no visible region absorption in the combination spectra (**1a** + **2k**) thus ruling out EDA mechanism (Scheme 6E). For details, see SI (Figure S18).

8. Please remove the arrow in the intermediate C because it is confusing with generation of intermediate B via SET process.

>> We have modified the color and style of the arrow in the intermediate C in the revised manuscript.

9. To justify only photocatalyst absorption of Blue LED absorption or exclude the EDA complex, authors should provide an absorption spectrum of all the starting materials or in combination.

>> We have added the UV-visible absorption spectra of photocatalyst, individual and combination of all starting materials. The individual substrates as well as their combination spectra (**1a** + **2k**) show no absorption in the visible region, thus excluding any possibility of EDA complex (Scheme 6E).

10. In mechanistic studies, one reaction needs to be conducted by using 1,6-diene as a radical acceptor.

>> We have conducted a reaction using dimethyl 2,2-diallylmalonate (1,6-diene) under the standard condition and afforded the cyclized product **25** in 63% yield (Scheme 6C). This experiment discloses strong support for the participation of malonyl radicals in this protocol via SET-type mechanism.

11. The authors proposed a reaction mechanism for the direct excitation of photocatalysts. The absorption spectra of the compound and the emission spectra of the light source should be reported in the main manuscript overexposed to demonstrate the possible direct excitation of the compound based on UV-vis spectra. And also move the light on-off experiments to the main manuscript.

>> We have taken note of the above suggestions and added the normalized absorption spectra of all starting materials and emission spectra of the 390 nm LED source in a single plot (Scheme 6E). These spectra show that starting materials don't absorb light in the 390 nm LED light source region. So the possibility of product formation via direct excitation of starting materials is not a viable pathway. The light on-off experiments have also been moved into the main manuscript (Scheme 6D).

12. Detailed mechanistic evidence is necessary to prove the reaction mechanism especially from intermediate C and E (I believe, this is a stable compound and isolable at low temperature or short reaction time).

>> We did isolate the intermediate **C** as **3a'** and already reported it in the original manuscript with **C** being labelled as a metastable compound. This may have been mistakenly overlooked. Further, we have tried to isolate the intermediate **E** (now re-numbered as **4'**), but this isn't possible even at low temperature or short reaction time as it automatically converted to **4** through a domino difunctionalization-isomerization-cyclization sequence. The spontaneous formation of **4a** from the reaction between **1a** and **2a** has already been pointed out during optimization study. As reaction of **1a** with **2a** under standard conditions provides a uncyclized **3a'** and cyclized **4a**, the study of reaction mechanism with these substate are problematic due to the presence of enolizable hydrogen on **2a**. To prove

the mechanism, we worked with a quaternary alkyl thiocyanate (**2k**) without enolizable hydrogen and provide all mechanistic studies using its thiocyanate product **5f** and isothiocyanate product **6f**.

13. To prove hypothesized reaction mechanism, authors should conduct DFT studies to define the key intermediates, Sulphur and nitrogen attack competing pathways.

>> Thank you for this suggestion. We have now conducted the DFT studies to strengthen our reaction mechanism in Scheme 7. The key intermediates have been defined previously with a series of details mechanistic studies as depicted in Scheme 6.

14. The presentation of schemes is congested in the manuscript. Please redraw or rearrange for a better presentation.

>> We are sorry about the congested presentation of the scheme in the original manuscript. Now we have separated each category of product (*S* and *N*) into individual schemes (Scheme 3 and 4) with other bifunctional chalcogenonitrile reagents in Scheme 5.

15. There are many typo and grammatical errors in the manuscript. The authors should revise it carefully. English polish is also necessary.

>> The manuscript was thoroughly checked to remove typos and grammatical errors. Also, the English of the manuscript has been polished.

16. Cite the following related papers on photoinduced reactions; a) ACS Org. Inorg. Au 2022, 2, 435–454; b) Nat. Commun., 2022, 13, 2345.

>> The aforementioned references have been added in the revised manuscript (ref. 18 and 42).

Supporting Information:

17. Some spectra feature very broad signals (**3g**, **3k**, **3i** and etc) in ¹H NMR. Please find this unusual outcome compared to other products in the manuscript.

>> We have reacquired the ¹H NMR spectra of **3g**, **3k**, **3i**, and a few other compounds. Now peak splitting was observed similar to the other products. We assume that instrumental spinning error may be responsible for these broad signals.

18. In the manuscript, the authors showed mixture of diastereomeric isomers (dr) in compounds **4n**. However, the ¹H and ¹³C NMR showed only single isomer. Please check it.

>> Thank you for pointing this out. The product **4n** (**4p**) was formed with the diastereomeric ratio (dr) 2:1, and we have presented the NMR data of major isomer only. [in bracket, new numbering]. We corrected the structure in the revised manuscript.

19. Few spectra are blurring (example; **3z**, provide high resolution spectra. In addition, some of the final products are not pure (for example: **5ad**, **5ac** and **5ab**). Repurify and recalculate the yields.

>> We respectfully disagree with this comment as **NO** blurry images in the spectra section were found upon rechecking and when we took all of them in the same way. Further, **we had NO compounds designated as 5ad, 5ac and 5ab in the entire manuscript that was submitted originally!** We strongly believe this comment has no relation to our manuscript at all and may have been mistakenly addressed to us, which is quite concerning.

20. Some ¹H NMR and ¹³C NMR spectra clearly show impurities (example **1ae**, **3y** and **7b** etc). Re-purification is required.

>> The above mentioned compounds as well as several others that were showing impurities have been repurified and spectral data has been reacquired.

21. The ¹³C spectra indication should be decimal, f.i. 177.6 instead of 177.57.

>> The issue has been corrected in the revised SI.

22. In Supporting Information, the author should include the solvent and NMR frequencies (MHz) in the copies of ¹H, ¹³C-NMR spectra.

>> This seems to be yet another instance of overlooking on the part of the reviewer as these informations have already been mentioned in the spectral data presented in the SI.

23. In compound **5d** spectra (¹H and ¹³C NMR), why authors give integration wrongly because there is a clear splitting in the peaks (for example at 3.0 ppm and 1.25 ppm). I believe this is because of the mixture of regio products (Sulphur and nitrogen attacked products). And also, the ¹³C NMR carbon count is too much. Please check it carefully in all the compounds.

>> This is not a mixture of regio-products but rather diastereomers. As the product **5d** (now as **3ah**) was derived from a mixed malonate ester, the ¹H and ¹³C NMR spectra contain a 1:1 diastereomeric mixture including rotamers of the compound. We have re-acquired and re-integrated the spectra in the revised manuscript.

Reviewer #3

Maity and co-workers have described the photocascade chemoselective controlling of ambident Thio(Seleno)cyanates via electron transfer. Overall, the work has been nicely explored the reactivity of thiocyanate by generating the thiocyanate radical. The novelty of this protocol lies in reductive cleavage of C-S bond via single electron transfer and isomerization of thiocyanate to isothiocyanate under photocatalytic reaction conditions. On the other hand, the addition of thiocyanate radical to alkene is well studied and the second acid catalysed cyclization step is also well known (Green Chem., 2015,17, 3515-3520; ACS Sustainable Chem. Eng. 2019, 7, 16, 14009–14015; Org. Biomol. Chem., 2019,17, 2232-2241). Importantly, the pathways of C-S bond cleavage and isomerization of isothiocyanate is still unclear. The CV and Stern-Volmer experiment are not sufficient enough to provide the insight into the C-S bond cleavage and ambident reactivity of thiocyanate. Based on these concerns and considering the lack significant novelty of the protocol, this reviewer found this manuscript not suitable for the publication in Nature communication.

>> We thank the reviewer for the valuable suggestions but respectfully beg to differ about the points on which this work is being compared with the above reports. In our opinion, this work, which provides a 2-component Markovnikov thiocyanation with novel bifunctionl thiocyanate group transfer reagent giving a number of new insights, upgrades as well as different reactivity (kindly refer to below scheme for an elaborative discussion) on chalcogenonitriles, compared to the commonly known protocols of 3-component *anti*-Markovnikov thiocyanation of aromatic olefins by oxidative generation of thiocyanate radical from thiocyanate salts. We have provided a detailed rebuttal of this point by including a scheme (VI and VII below) to underpin the difference, and articulated about the novelty of the reported concept.

VI. 3-Component thiocyanation of alkenes:

Oxidative generation of thiocyanate radical from thiocyanate salt (*common known reactivity*)

VII. 2-Component Group Transfer thiocyanation and isothiocyanation from alkylthiocyanate bi-functional reagent (*Unknown, this work*)

While we agree that the hetero-nucleophilic cyclization at the electrophilic nitrile centre of –SCN is quite feasible as referred to in the mentioned papers, Markonikov -carbothiocyanation and -isothiocyanation of alkene by a redox neutral pathway using a single alkylthiocyanates reagent does not exist anywhere in the literature. The known and literature-enriched thiocyanation of vinyl arenes are commonly initiated by the SCN[•] radical generated by oxidation of inorganic thiocyanate salt (Scheme 1C). Here we have not only introduced various alkylthiocyanates as new bi-functional reagents under photoredox conditions for redox-neutral transfer of -SCN and -NCS groups but also developed new catalysts which can selectively install those groups onto alkenes in regio- and chemo-selective manner by a control of the redox potential (Scheme 1D). The working principle of this process is controlled by a catalytic-switch which regulates the desired product through cascade catalysis. The mechanism of this cascade process involves a selective combination of a kinetically fast radical-chain process (for-SCN product) and a slow radical-polar process (for-NCS product) working sequentially aided by a photocatalyst. Hence, the developed method and in particular the catalyst-regulated controlling of ambident-SCN, is quite unique. Moreover, the higher chalcogenonitrile, -SeCN have also been controlled in the same fashion to install the selenocyanate and iso-selenocyanate groups onto alkenes. Pleasingly, further cyclization of difunctionalization products directly allows us the access to *S*-, *Se*-, *N*-heterocycles in step-, atom- and redox-economic fashion from feedstock alkenes by a bimolecular formal [2+2+1] hetero-annulation. We believe that the developed manipulation strategy will help organic chemists find new applications with the novel chalcogenonitrile group transfer bi-functional reagent, inorganic chemists to design chemo-divergent ligands [-S(Se)CN, -NCS(Se)] and medicinal chemists to access library of divergent heterocycles from same starting pool.

>> Regarding the detailed studies on the reaction mechanism:

We have added thorough mechanistic studies in Scheme 6 and their conclusions in the subsequent parts of the revised manuscript (further details are provided in the SI). We hope this will clear the doubts regarding the mode of C-S bond cleavage as well as the isomerization involved in the proposed cascade catalysis. Additionally, We have now conducted DFT calculations to provide theoretical evidence for strengthening our proposed mechanism and presented their conclusions in Scheme 7. A summary of the detailed mechanistic studies are as follows:

Indicates the reaction involves a radical pathway

Malonyl radical is the key reaction intermediate

D' indicates no direct excitation of substrates and no EDA complex formation

E' proves light necessary for propagation of reaction

F' indicates both light and catalyst are necessary for the isomerization step

G' provides evidence for carbocation formation during isomerization

H' indicates initially (within 20 min) 5f is the major product and 6f is minor. Subsequently 6f increases as 5f is gradually converted into it, demonstrating a cascade catalytic reaction where the isomerization is the rate-limiting step.

I' shows PTH-4 is suitable for both 1st and 2nd step as it is quenched by 2k and 5f both. Whereas, PTH-2 is selective for the 1st step, only being quenched by 2k.

J' indicates SET is the preferred pathway over EnT as only catalysts with potentials high enough to reduce the 2k and 5f are able to generate products despite all of them having higher triplet energy than the substrates.

Fig: Various investigations conducted for deciphering the plausible mechanism

1. Scheme present in the table-1 is not appropriate, the product presented in the table 1 reaction is the product of the second step and the condition of the second step is not presented in the reaction. The final product formations happen in two steps and it is not depicted in the reaction of table 1 second step reaction condition needed to be included.

>> We have added the conditions for the cyclization step as well in the scheme presented in Table 1. It is to be noted that while the *S*-cyclization does need Lewis acid treatment, *N*-cyclization product is directly obtained from alkene and thiocyanomalonate under photoredox conditions, needing no Lewis acids. To compare the yield of both the cyclized products in the reaction, the crude mixture was as such treated with AlCl₃ and submitted for ¹H NMR with internal standard. Thus, while *S*-cyclization is indeed a 2-step process, *N*-cyclization is rather a domino sequence of difunctionalization-isomerization-cyclization culminating in 2-thiopyrroldones in one go.

2. Table-1, entry 14, indicating that the final product formation (2%) is just in 30 min irradiation of 390 nm light without photocatalyst. However, the author did not present the same reaction data after long reaction time such as after 24/48 h. This 2% yield of the final product in just 30 min suggests us to think critically about C-S bond cleavage of compound 2. Is this C-S bond cleavage because of the 390 nm light irradiation.? or is it the heat generated because of the 390 nm light irradiation inside the reaction responsible for such C-S bond cleavage.? Control experiments need to be performed to address this product formation such as; same reaction for 24 h or 48 h without photocatalyst with 390 nm light and secondly the reaction at higher temperature 60 oC without irradiation of light with and without photocatalyst. (report based on C-S bond cleavage (α - to carbonyl) by simple irradiation of light - Am. Chem. Soc. 2017, 139, 40, 14315–14321; Chem. Commun., 2023, 59, 5343)

>> Thank you very much for this valuable suggestion. Irradiation upto 24 h led to the desired **3a** product being detected in ¹H NMR in only 2-4% yield (entries 14 and 15, Table 1). This suggests that trace amount of C-S bond cleavage does take place in compound **2** which may be due to 390 nm light being used for irradiation, however it is not a feasible mode as there is no considerable improvement in yield even after a long run. The reaction temperature of our setup usually does not exceed 32 °C as it is being cooled by a high-speed fan (see SI for reaction setup with temperature detector). Upon carrying out the reaction at an increased temperature of 60 °C in the absence of light and with/without photo-catalyst, the yield was still in traces (entries 17 and 18, Table 1) which suggests that whatever heat is generated during irradiation has no influence on the reaction outcome.

3. Is this C-S bond cleavage because of the single electron reduction.? or is it the energy transfer from the catalyst to molecule 2 triggering a homolytical bond cleavage resulting in two stable radical species? This aspect has not been considered, this ambiguity of energy transfer or electron transfer needs to be clarified. This reviewer suspects that one of the products formation could be because of the energy transfer because the thiocyanation and isothiocyanation are known to be kinetically and thermodynamically controlled.

>> We thank the reviewer for this valuable insight. To gain a better understanding about the energy-transfer mode of catalyst activation for C-S bond cleavage, we have computationally calculated the triplet energy (E_T) of both substrates **2k**, **5f** and catalysts **PTH2**, **PTH4**, *fac*-Ir(**ppy**)₃. Additionally, the electrochemical measurements ($E_{1/2}$) of corresponding substrates and catalyst were also done. Then the possibility of both EnT and SET were examined extensively by carrying out each of the steps of the cascade reaction individually with a series of catalysts that have been traditionally known for energy transfer, along with our initial SET capable catalysts. Even though all the catalysts in the series had triplet energies higher (>45 Kcal/mol) than the starting thiocyanates **2k**

(44.9 kcal/mol) and **5f** (43.6 kcal/mol) thus making them capable of driving the reaction by C-S bond homolysis, only those catalysts with reduction potential higher than the substrates **2k** ($E^{\text{red}} = -1.36$ V vs SCE, entries 2, 5-8) and **5f** ($E^{\text{red}} = -1.83$ V vs SCE, see SI for Electrochemical Measurements), were productive (entries 5 and 8). This proved that triplet energies and by inference the EnT route had no role to play in this cascade, with SET being the plausible pathway.

Investigation into catalyst activation modes

entry	photocatalyst (x mol %)	E_T (kcal/mol)	$E_{1/2}$ (M^+/M') V vs. SCE	Light (nm)	ATRA output		Isomerization output
					t (h)	5f/6f (%) ^b	6f (%) ^b
1	Benzophenone (5)	69.1	-0.61	370	6	4/0	3
2	3DPA2FBN (5)	65.5	-1.60	390	6	32/3	2
3	Thioxanthone (5)	63.4	-1.11	390	6	3/0	3
4	Ir(dFCF ₃ ppy) ₂ (dtbpy)PF ₆ (1)	61.8	-0.89	450	6	6/0	trace
5	PTH4 (5)	58.4 ^a	-2.16	390	0.5	57/32	85
6	PTH2 (5)	58.1 ^a	-1.53	390	0.5	87/5	5
7	fac-Ir(ppy) ₃ (1)	58.1(57.9) ^a	-1.73	450	6	68/5	6
8	PTH3 (5)	55.3	-2.10	390	0.5	51/30	81

^a computationally calculated value. ^b crude ¹H NMR yield.

The discussion has been added in the revised manuscript (mechanistic studies) and the table in Scheme 6J.

4. This protocol is not explored for the electron deficient double bond except example **3y** more such examples need to be explored.

>> We have taken note of this and conducted the reaction of electron-deficient alkenes with our developed methods, obtaining the desired product with moderate yields. We have summarized the isolated product with the yields for the respective alkenes.

Used alkenes	Reaction with PTH2
1ab	3aa (39%)
1ac	3ab (41%)
1ae	3ad (42%)

5. NMR spectra of some compounds are not pure such as **3a'**, **3c**, **3i**, **3w** and **9a**.

>> We have addressed the issue by replacing the ^1H , ^{13}C , and ^{19}F NMR spectra of **3a'**, **3c**, **3i**, **3w** (**3y**), **9a** (**7a**) as well as several other compounds. [brackets () indicate new numbering in revised manuscript].

REVIEWER COMMENTS

Reviewer #1 (Remarks to the Author):

In this manuscript, Maity and co-workers have revised the corresponding text in the revised manuscript and SI to address most of my concerns involving the generality of substrate scope, key mechanistic studies, and other details. Given the importance and novelty of this work, I still recommend publication in Nature Communications after minor revision: please check and revise the structure of 4m in the revised SI (S219).

Reviewer #2 (Remarks to the Author):

[Note from the Editor: Reviewer #2 was asked to look over the response given to reviewer #3 who was not available to assess the revision.]

The authors have addressed all my concerns, regarding mechanistic study, substrates scope and other queries in the response letter with corresponding experimental results. These newly added results into the manuscript and supplementary information could provide a better understanding of this advanced methodology. As I mentioned in the previous comments, I won't agree on the true novelty of this methodology compared to existing literature data (Tetrahedron Letters, 1987, 28, 117-120; Org. Lett. 2023, 25, 3564–3567; American Chemical Journal, 1901, 26, 353; Org. Biomol. Chem., 2008, 6, 2995–2999; J. Am. Chem. Soc., 1988, 10, 8679-8685; J. Serb. Chem. Soc., 2015, 80, 453–458; J. Org. Chem., 1993, 58, 3355-3360). However, based on the authors exceptional additional experimental results, selectivity, broadness proved their hypothesis shows an advancement compared to existing literature data. Therefore, I would like to recommend this work in “nature communications” after major revision noted below.

Detailed comments:

1. The authors must expand the scope of 2o and diverse alkenes derivatives to showcase the broadness of this synthetic methodology similar to Scheme 3 and 4.
2. Why do the authors pick 2k as a radical precursor instead of 2a in Scheme 6 mechanistic studies? Because their main concept is to build products 3 and 4.
3. In this work, the authors synthesized thiocyanatomalonates from the corresponding malonates by a one-pot process involving bromination, followed by thiocyanation. I will suggest trying the reaction

by using diethyl 2-bromomalonate and ammonium thiocyanate with alkene under standard conditions (For example; Org. Lett. 2023, 25, 3564–3567). If this works, it will be interesting and improve the novelty of the reaction.

4. Few compound HRMS analysis is not within the 10-ppm acceptable error (for example: 3a, 3c, 3t, 3z etc.). Please recheck in the entire SI and provide the original HRMS spectra for review.

5. The authors showing compound 3ah, 5i dr ratio is 1:1 but the respective NMR spectra are not in 1:1 ratio. On what basis the authors given this ratio? It is completely misleading the readers.

6. Still some NMR spectra of compounds are not pure such as 4ad, 4ag, 5a, 5d, 5e, 6d, 6i, 7a, 8b and 9.

7. Still there are many typos and grammatical errors in the manuscript and supporting information. Few things are noticed below.

a) In page 10, the color coating of the compound 9 is different with SI. Please maintain consistency.

b) The “mol %” should be revised to “mol%” in the manuscript and SI where it is necessary.

c) Please write the respective solvent quantity (page S21; Solvent (x M)).

Nature Communications R2 comments

The authors have clearly done a major effort in revising their original manuscript, including the mechanistic investigation with dispersion-corrected density functional theory. This, together with a more logical and consistent build-up of the supporting information has added significantly to the scientific merit of the manuscript. I really appreciate their hard work and efforts to improve the scientific quality and quantity for the broad readership nature communications journal. But, I strongly reject the author's claims this work is a novel, alternatively suggest they to claim advancements in the addition reactions with new thio(seleno)cyanates. If they have done this work with other radical acceptors, I will agree with their claims but they utilized well-known radical acceptors (alkenes) with modified substituted thio(seleno)cyanates on active methylene carbon under metal free photocatalysis for the synthesis of new heterocyclic in a (-SCN vs -NCS) chemoselective approach and also chemoselective 1,2-addition products. I agree there are few new and interesting steps in the present methodology but previously the active methylene carbon malonate derivatives utilization (-SePh, -TEMPO, -halo) (*J. Org. Chem.* 1993, 58,3355-3360; Sunlamp photolysis without external photocatalyst) has been exploited in radical reactions with alkenes and alkynes (both terminal and internal) without photocatalyst. For the authors notice, the radical generation from altered precursor, radical addition on alkenes are common and they cannot include their claim on this portion of work is as novel. Moreover, the reviewer-3 also pointed out, the addition of thiocyanate radical to alkene is well studied and the second acid catalyzed cyclization step is also well known. According to my knowledge, the new advancements in the present work is chemoselective addition (-SCN vs -NCS) in the presence of metal-free photocatalyst (PTH4) which dictates cyclization for the product **4** formation. Based on the above advancements, extensive additional experiments in the manuscript along with detailed mechanistic studies, broad substrate scope, diverse new scaffolds, mild conditions and improved Supporting Information, I would like to recommend this advanced manuscript for publication in the prestigious journal nature communications after major revision noted below.

Some detailed comments:

1. The way the entire manuscript presented still lacks the necessary thoroughness. Please make it easy to understand for the broad scientific community readers. What's the author's problem

to expose and showcase their work with excellent presentation skill of the Schemes? They are still congested in Scheme 1, Scheme 2 and Scheme 5. Please improve presentation of the manuscript.

2. What about the stability of these new types of substituted scaffolds (**3** and **4**)? Is it stable at room temperature or need to store at cooler temperature for a long life? Need to include few sentences about it in the manuscript.
3. Most importantly the authors mentioned both products (**3b** and **4b**) isolated silica gel column chromatography purification at same Rf (**3b**; Rf value = 0.4 [EtOAc:Petroleum ether = 1:4 (v/v)] and **4b**; Rf value = 0.4 [EtOAc:Petroleum ether = 1:4 (v/v)]). If this is true, how did the authors get pure **3b**? Because synthesis of **3b** can also produce the minor product as a **4b** in the same transformation. (According to optimization table-1, entry 2, a still minor product is forming along with the desired product).
4. For the better understanding, please provide a TLC (thin layer chromatography) progress monitoring pictures for entry 2 and entry 10 in Table 1. For entry 10, the authors should provide (TLC pictures) different intervals of the reaction time (1 h, 2 h, 4 h, 8 h, 12 h, 16 h, 20 h, 24 h). If the authors include these pictures in the Supporting Information, it will be more useful for the readers (synthesis of **3** and **4**).
5. In Table-1 and Table-S1, there is no protic solvent optimization in the reaction. Is there any reason or observed different products?
6. All the reactions were conducted under an inert atmosphere. Are there any side products observed under oxygen? Please include the observation in the manuscript.
7. On what basis, the author's claimed this protocol is an atom-economic? In most of the cases the radical acceptors (alkenes) are used in excess amounts. Where is the mass balance from starting material to product formation?
8. Please provide a scientific explanation for lower yields in the few final products (**3o**, **3n**, **4m**, **4n**, **4w**). In electron-rich alkenes they observed side product **28** in the **4r** synthesis and there is no any discussion in the manuscript text but they provided characterization of the compound in the SI. As an honest synthetic chemistry, the authors must present their negative results also in the manuscript especially Table S6.

9. The authors claimed the final step is isomerization for product 4 formation. To find the exact source of proton (N-H), the authors should synthesize the deuterium incorporated carbon of the starting material **2** and test under standard reaction conditions.
10. I am curious, the authors performed a reaction in the presence of nucleophilic solvent Toluene. Did they observe any side product formation with toluene involvement in the transformation? Because in the Scheme 6G isomerization of the intermediate evaluation **5f**, the nucleophiles (EtOH and 1,2,4-trimethoxy benzene) are trapped with an in situ generated carbocation intermediate (**5f'**).
11. It will be interesting, if the authors expand the scope of the compounds **7a** and **8a** to other derivatives.
12. Room temperature can vary widely with climate, season, and time of day. To clearly state the conditions of the study in a manner that facilitates replication by other researchers, please consider using an approximate temperature range instead.
13. In most of the cases the products yield given in only Schemes. Please mention the yields of the compounds in the manuscript text discussion (either specific number or range).
14. Previously the authors selected internal alkenes such as 1,2-dihydronaphthalene, indene, norbornene but I won't agree with their internal alkene's examples because these examples will not expose the regioselectivity problems in the present methodology. Please test the following as a starting material.

Miscellaneous in the manuscript:

- a) In Scheme 5, the authors should provide detailed stoichiometric amounts of starting material, reagents, exact reaction time and other information.
- b) Please provide the exact quantity of the gram-scale experiment conducted and obtained yields in the manuscript text.
- c) The authors observed good yields with 100% intensity of LEDs. Please include it in the entire manuscript schemes and footnotes.

- d)** The compound number **27** is appears in the manuscript but no respective structure. Please provide it.
- e)** The word “isomerization cycle” given in the photocatalyst cycle. I think remove from there and include in the respective place (the readers will confuse the photocatalyst may isomerize).
- f)** Same number (**3ac**) is given for different structures. Please see in Scheme 3 and Scheme 4.
- g)** In Scheme, for compound **23** formation, the authors mentioned dr ratio is 1:1. That is not a dr mixture, revise to *E/Z* mixture ratio in the manuscript and SI.

Supporting Information:

15. In SI, still a lot of mistakes and typo errors are present. Few are noticed below. The authors must take extra care to resolve these issues in the revision.

- a)** How is it possible to multiplet (1.33 – 1.33 (m, 6H)) for methyl groups in the compound **3l**? Same issue in the other compounds 3m, 3o, 3q, 3u, 3v, 3w (few I given here, the authors should revise where it is necessary in the other compound).
- b)** 7.39 (t, $J = 7.9$ Hz, 2H) in 3ab, which proton peak will split the triplet?
- c)** 2.97 (brs, 1H) in 3ae, which proton peak will belong to the broad singlet?
- d)** 2 7.23 – 7.17 (m, 4H) given multiplet for compound 5h, how it is possible because there are 2 H is same environment and other 2 H is different environment.
- e)** The HRMS analysis is not within the 10 ppm (HRMS (ESI) m/z calcd for $C_{18}H_{27}N_2O_4S$ $[M+NH_4]^+$: 367.1692; found: 367.1674) acceptable error for compound **5f**. Please provide HRMS spectra. Need to check carefully other compounds.
- f)** For compound 17, authors mentioned (for the mixture) mixtures in the characterization, can you tell us what mixtures they are?
- e)** For compounds **5i** and **6i** given a mixture of dr ratio (1:1) but this is not correct ratio based on 1H NMR and ^{19}F NMR. This is a clear misleading of the readers. Please check other diastereomers compounds.

There are many incorrect integrations of the peaks, coupling constants in the compound characterization and their respective compounds 1H NMR spectra. I strongly suggest them to do it carefully in the revision.

16. In the SI, some of the final compounds (3d, 3e, 3k, 3o, 3ae, 3af, 4r, 4x, 4ac and 5g etc.) still show solvent peaks or impurities in the spectra. The authors must provide pure spectra in the revision.

Response:

Reviewer #1

In this manuscript, Maity and co-workers have revised the corresponding text in the revised manuscript and SI to address most of my concerns involving the generality of substrate scope, key mechanistic studies, and other details. Given the importance and novelty of this work, I still recommend publication in Nature Communications after minor revision: please check and revise the structure of 4m in the revised SI (S219).

>> We thank the reviewer for finding our work suitable for publication.

We corrected the structure of **4m** in the revised SI (S234).

Reviewer #2

The authors have clearly done a major effort in revising their original manuscript, including the mechanistic investigation with dispersion-corrected density functional theory. This, together with a more logical and consistent build-up of the supporting information has added significantly to the scientific merit of the manuscript. I really appreciate their hard work and efforts to improve the scientific quality and quantity for the broad readership nature communications journal. Based on the above advancements, extensive additional experiments in the manuscript along with detailed mechanistic studies, broad substrate scope, diverse new scaffolds, mild conditions and improved Supporting Information, I would like to recommend this advanced manuscript for publication in the prestigious journal nature communications after major revision noted below.

>> We thank the reviewer for finding our work suitable for publication.

Some detailed comments:

1. The way the entire manuscript presented still lacks the necessary thoroughness. Please make it easy to understand for the broad scientific community readers. What's the author's problem to expose and showcase their work with excellent presentation skill of the Schemes? They are still congested in Scheme 1, Scheme 2 and Scheme 5. Please improve presentation of the manuscript.

>> We are thankful for the comment. We have thoroughly checked and improved the presentation throughout the manuscript and made these schemes (Fig. 1, Fig. 2, and Fig. 5) more spacious to make it easy to understand for the broader scientific community. Additionally, we have polished the manuscript's language to make it easier to understand.

2. What about the stability of these new types of substituted scaffolds (**3** and **4**)? Is it stable at room temperature or need to store at cooler temperature for a long life? Need to include few sentences about it in the manuscript.

>> We thank the reviewer for the comment. The substituted thiopyrrolidone scaffolds **4** are generally stable at room temperature, but substituted 2-imino-tetrahydrothiophene scaffolds **3** need to be stored in the refrigerator. In the selenium series, selenopyrrolidone derivatives **8** are also stable, but 2-imino-tetrahydroselenophene derivatives

7 are not bench stable and require a cooler temperature for storage. We have addressed the concerns about their stability in the revised manuscript: “The newly synthesized N-heterocyclic scaffolds (both thiopyrrolidones **4** and selenopyrrolidones **8**) are normally stable at room temperature. However, their substituted hydrothiophene **3** and hydroselenophene **7** counterparts are not bench stable and require cooler storage temperatures.”

3. Most importantly the authors mentioned both products (**3b** and **4b**) isolated silica gel column chromatography purification at same R_f (**3b**; R_f value = 0.4 [EtOAc:Petroleum ether = 1:4 (v/v)] and **4b**; R_f value = 0.4 [EtOAc:Petroleum ether = 1:4 (v/v)]). If this is true, how did the authors get pure **3b**? Because synthesis of **3b** can also produce the minor product as a **4b** in the same transformation. (According to optimization table-1, entry 2, a still minor product is forming along with the desired product).

>> We thank the reviewer for asking us for clarification. Initially, we considered a single decimal place for the presentation of the R_f value of a compound, but now we changed it to two decimal places to differentiate the exact position of the compounds in a mixture in TLC. Now the R_f of **3b** and **4b** appear as 0.37 and 0.42 (picture of TLC attached). We have now modified the R_f values of all compounds to two decimal places in the entire SI.

S = spot of compound 3b CO = spot of compounds 3b and 4b N = spot of compound 4b R_f value of 3b = 0.37 [EtOAc:Petroleum ether = 1:4 (v/v)]. R_f value of 4b = 0.42 [EtOAc:Petroleum ether = 1:4 (v/v)]. TLC staining: iodine vapor	--	--

4. For the better understanding, please provide a TLC (thin layer chromatography) progress monitoring pictures for entry 2 and entry 10 in Table 1. For entry 10, the authors should provide (TLC pictures) different intervals of the reaction time (1 h, 2 h, 4 h, 8 h, 12 h, 16 h, 20 h, 24 h). If the authors include these pictures in the Supporting Information, it will be more useful for the readers (synthesis of **3** and **4**).

>> As suggested, we have added the TLC pictures of entry 2 and entry 10 of Table 1 (entry 2 and entry 26 in Supplementary Table 1) in the revised SI (Page S21). We have also included the TLC picture of reaction progress of entry 10, Table 1 (entry 26 in Supplementary Table 1) with different intervals of the reaction time (1 h, 2 h, 4 h, 8 h, 12 h, 16 h, 20 h, and 24 h) in the revised SI (Page S21).

5. In Table-1 and Table-S1, there is no protic solvent optimization in the reaction. Is there any reason or observed different products?

>> We thank the reviewer for the observation. We have conducted the standard reaction in ethanol and products **3a/4a** were obtained as 63% : 4%. We have included this solvent in the optimization table in SI (entry 14, Supplementary Table 1).

6. All the reactions were conducted under an inert atmosphere. Are there any side products observed under oxygen? Please include the observation in the manuscript.

>> We thank the reviewer for the comment. We have previously monitored the negative impact of aerial oxygen on reaction yield (78% vs. 89%) in SI (entry 17 vs. 2, Supplementary Table 1). We further repeated the standard reaction under open air but didn't identify any side product. The impact of aerial oxygen on reaction output was now added to the manuscript (entry 17, Table 1) with a line: *"Aerial oxygen has a negative impact on the reaction yield (entry 17, table 1)."*

7. On what basis, the author's claimed this protocol is an atom-economic? In most of the cases the radical acceptors (alkenes) are used in excess amounts. Where is the mass balance from starting material to product formation?

>> As such, the total formula mass of atoms in the desired product is the sum of the formula mass of atoms in the starting materials- a transformation that maintains the atom economy. But, yes, as the radical acceptors were used in excess- the reaction is not mass-balanced. Hence, we agree that the term "atom-economic" may not be entirely suitable in the present case. To this end, we have refrained from using it in the revised manuscript.

8. Please provide a scientific explanation for lower yields in the few final products (**3o**, **3n**, **4m**, **4n**, **4w**). In electron-rich alkenes they observed side product **28** in the **4r** synthesis and there is no any discussion in the manuscript text but they provided characterization of the compound in the SI. As an honest synthetic chemistry, the authors must present their negative results also in the manuscript especially Table S6.

>> The products **3n-3o** and **4m-4n** are obtained in lower yields due to the deterioration of the parent alkenes under the reaction conditions. The product **4w** (**4x**) is obtained in diminished yield due to the polymerization of the parent alkene. This is the case with some other related compounds as well and a brief discussion on this has been included in the revised manuscript. Meanwhile, the byproduct **28** (**4s'**) is obtained as a result of the formation of the

α -ethoxy carbocation in the oxidative radical polar crossover step and this has been discussed in the revised manuscript as well along with its structure (Fig. 4): “In case of **4s**, considerable amount of aldehyde **4s'** is formed possibly due to the generation of α -ethoxy carbocation during the oxidative polar crossover step.⁴⁷”. Additionally, the negative results of some of the substrates were now discussed in the appropriate position in the manuscript.

9. The authors claimed the final step is isomerization for product **4** formation. To find the exact source of proton (N-H), the authors should synthesize the deuterium incorporated carbon of the starting material **2** and test under standard reaction conditions.

>> We are thankful for the suggestion. For product **4** formation, the 2nd step is a cascade process involving photoredox isomerization (**3'** \rightarrow **4'**, Fig. 2A) followed by spontaneous cyclization (**4'** \rightarrow **4**, Fig. 2A). As suggested, we have synthesized the deuterium incorporated thiocyanatomalonate **2a-d** (73% D) and conducted the reaction with 4-methylstyrene (**1a**) under standard conditions. The crude ¹H NMR analysis indicates the formation of deuterated product **4a-d** with 58% deuterium incorporation. This result indicates the primary source of proton (N-H) in product **4** is the methine proton of the starting thiocyanomalonate **2a**.

The results have been incorporated in Fig. 6C in the manuscript and in SI (page: S74). A brief discussion was also added in the manuscript: “To understand the source of the N-H proton in **4a**, deuterated-TCM **2a-d** (73% D) was synthesized and reacted with **1a** under standard conditions with **PTH4**. Product **4a-d** was detected in crude ¹H NMR with 58% deuterium incorporation, indicating the methine hydrogen of **2a** is mainly supplying the N-H proton of product **4a** (Fig. 6C).”

10. I am curious, the authors performed a reaction in the presence of nucleophilic solvent Toluene. Did they observe any side product formation with toluene involvement in the transformation? Because in the Scheme 6G isomerization of the intermediate evaluation **5f**, the nucleophiles (EtOH and 1,2,4-trimethoxy benzene) are trapped with an in situ generated carbocation intermediate (**5f'**).

>> We thank the reviewer for this curiosity. As suggested by the reviewer, the crude mixture of the reaction between **1a** and **2a** conducted in toluene solvent was analyzed by both ¹H NMR and HRMS. However, we didn't find any corresponding spectral data which should be associated with the toluene adduct.

11. It will be interesting, if the authors expand the scope of the compounds **7a** and **8a** to other derivatives.

>> As suggested, we have further expanded the scope of the compounds **7a** and **8a** to other derivatives to showcase the broadness of this synthetic method. As shown below, the methodology worked well with various alkenes and selenocyanatomalonates to obtain corresponding Se- and N-heterocycles with moderate to good yields (44-72%) under the optimized conditions.

The results have been added to the lower part of Table 2 and have been discussed in the manuscript as well. The related spectra have also been added in SI (page no: S281-S285; S288-S292).

12. Room temperature can vary widely with climate, season, and time of day. To clearly state the conditions of the study in a manner that facilitates replication by other researchers, please consider using an approximate temperature range instead.

>> Now, we replaced the rt with a range of 30-35 °C in all the schemes and footnotes in the manuscript. Additionally, we added a line at the beginning of the SI under general considerations: “All reactions conducted at rt refer to the temperature range of 30-35 °C.”

13. In most of the cases the products yield given in only Schemes. Please mention the yields of the compounds in the manuscript text discussion (either specific number or range).

>> As advised, we have included the range of yields obtained for the compounds in the manuscript text.

14. Previously the authors selected internal alkenes such as 1,2-dihydronaphthalene, indene, norbornene but I won't agree with their internal alkene's examples because these examples will not expose the regioselectivity problems in the present methodology. Please test the following as a starting material.

>> We thank the reviewer for the suggestion. We have conducted the reaction with the mentioned alkenes using our developed methods and summarized the results below.

alkenes		Reaction with PTH2	Reaction with PTH4
 1q		 3q, 58% (dr 1.4:1)	 4p, 51% (dr 5:1)
		No reaction takes place with diethyl 2-thiocyanatomalonate (2a). It may be due to the steric repulsion by bulky aryl groups of alkenes to the malonyl radical addition. We have added this in the manuscript: “Unfortunately, triphenylethylene and stilbene were unreactive, likely because of steric hindrance.”	
 1ac		 3ab, 65% (dr 4:1)	Not applicable. As illustrated in Fig. 7, isomerization of –SCN to –NCS in 2 nd catalytic cycle involves carbocation intermediate, generated by the oxidation of carbon-centered radical through the radical-polar-crossover mechanism. However, alkyl radical oxidation after the addition of malonyl radical to aliphatic alkenes is relatively difficult, and reaction with aliphatic alkenes leads to thiocyanates/S-cyclization products.
 R = aliphatic	 1ab Substrate 1ab was already there in the original manuscript!	 3aa, 67%	
	 1aa	 3z, 72%	

We included these new structures in Fig. 3 and Fig. 4 as appropriate.

Miscellaneous in the manuscript:

a) In Scheme 5, the authors should provide detailed stoichiometric amounts of starting material, reagents, exact reaction time and other information.

>> As suggested, Scheme 5 (Fig. 5B) has been modified with the detailed reaction conditions with other necessary information.

b) Please provide the exact quantity of the gram-scale experiment conducted and obtained yields in the manuscript text.

>> As suggested, we have provided the necessary information in the manuscript text as follows: *“The reaction between diethyl thiocyanomalonate 2a and 4-methylstyrene 1a was accomplished on a 6 mmol scale affording 3a (1.63 g, 81%, Fig. 3) and 4a (1.45 g, 72%, Fig. 4) under conditions A and B respectively, demonstrating the robustness of the process.”*

c) The authors observed good yields with 100% intensity of LEDs. Please include it in the entire manuscript schemes and footnotes.

>> We thank the reviewer for the comment. We have included the intensity of LEDs in the entire manuscript's schemes and footnotes. Additionally, we added a line at the beginning of the SI under general considerations (photoreaction): *“Unless otherwise noted, all reactions were conducted with 100% light intensity using Kessil light.”*

d) The compound number **27** is appears in the manuscript but no respective structure. Please provide it.

>> As advised, we have added the structure of compound **27** in Fig. 6G.

e) The word “isomerization cycle” given in the photocatalyst cycle. I think remove from there and include in the respective place (the readers will confuse the photocatalyst may isomerize).

>> As suggested, we have removed the phrase “isomerization cycle” from the photocatalytic cycle in Fig. 7A.

f) Same number (**3ac**) is given for different structures. Please see in Scheme 3 and Scheme 4.

>> We have corrected the compound numbering as **4ad** in Fig. 4.

g) In Scheme, for compound **23** formation, the authors mentioned dr ratio is 1:1. That is not a dr mixture, revise to *E/Z* mixture ratio in the manuscript and SI.

>> We revised and re-labeled it as *E/Z* isomers in the revised manuscript and SI.

Supporting Information:

15. In SI, still a lot of mistakes and typo errors are present. Few are noticed below. The authors must take extra care to resolve these issues in the revision.

>> We have revised the supplementary information thoroughly to rectify these errors.

a) How is it possible to multiplet (1.33 – 1.33 (m, 6H)) for methyl groups in the compound **3l**? Same issue in the other compounds **3m**, **3o**, **3q**, **3u**, **3v**, **3w** (few I given here, the authors should revise where it is necessary in the other compound).

>> We thank the reviewer for the observation. We have revised the said data as “1.31 (t, *J* = 7.1 Hz, 3H), 1.28 (t, *J* = 7.1 Hz, 3H).” We have also rectified this issue in the spectral data for the rest of the compounds wherever necessary.

b) 7.39 (t, *J* = 7.9 Hz, 2H) in **3ab**, which proton peak will split the triplet?

>> The marked (Red color) protons of compound **3ab** (**3ae**) are responsible for this peak. This peak has the appearance of a general triplet, however, the peak intensities of the individual splittings don't satisfy the conditions to be called a triplet. Hence, we have corrected this peak as “7.41 – 7.37 (m, 2H)” in the revised SI. [brackets () indicate new numbering in revised manuscript].

c) 2.97 (brs, 1H) in 3ae, which proton peak will belong to the broad singlet?

>> We are thankful for the observation. We have reacquired the ¹H NMR spectra of **3ae** (**3ah**) and **3m**, with peak splitting being clearly observable in a similar fashion to the other products of the series. We assume instrumental spinning error may be responsible for the unsatisfactory splitting of the said peak.

d) 2 7.23 – 7.17 (m, 4H) given multiplet for compound 5h, how it is possible because there are 2 H is same environment and other 2 H is different environment.

>> We have made necessary changes to the integration in the spectrum of compound **5h**, and it has been rewritten as “7.22 (d, *J* = 8.4 Hz, 2H), 7.18 (d, *J* = 8.3 Hz, 2H)” in the revised SI. Spectral data of other compounds with the same issue have also been rectified accordingly.

e) The HRMS analysis is not within the 10 ppm (HRMS (ESI) *m/z* calcd for C₁₈H₂₇N₂O₄S [M+NH₄]⁺: 367.1692; found: 367.1674) acceptable error for compound **5f**. Please provide HRMS spectra. Need to check carefully other compounds.

>> We thank the reviewer. However, to the contrary, the guidelines of the journal dictates the acceptable range of the HRMS analysis to be within 0.003 (i.e., 30 ppm) for molecules with *m/z* < 1,000. Nevertheless, we have appended a reacquired HRMS spectra of the said compound, which is more accurate with a [M+Na] peak within 5 ppm. The original spectra are also attached for reference.

Mass spectrometry

Authors should also provide mass spectral data to support molecular weight identity. High-resolution mass spectral (HRMS) data are preferred, and when *m/z* < 1,000, the calculated and found values should be within 0.003. When *m/z* > 1000, we expect an experimental value within 1 ppm of the calculated value.

HRMS of compound **5f**: (*old*)

HRMS of compound **5f**: (*reacquired*)

f) For compound 17, authors mentioned (for the mixture) mixtures in the characterization, can you tell us what mixtures they are?

>> We apologize for this inadvertent error. To address the issue, we have re-synthesized compound 17 using a different method (*Adv. Synth. Catal.*, **2022**, *364*, 689-694), leading to its isolation as a single compound as expected. The revised procedure, characterization data, and spectra have been added in SI (pages S66 and S304).

e) For compounds 5i and 6i given a mixture of dr ratio (1:1) but this is not correct ratio based on ¹H NMR and ¹⁹F NMR. This is a clear misguiding of the readers. Please check other diastereomers compounds.

There are many incorrect integrations of the peaks, coupling constants in the compound characterization and their respective compounds ¹H NMR spectra. I strongly suggest them to do it carefully in the revision.

>> We are thankful for the comment. We have corrected the dr ratio (dr 1.2:1 for 5i and dr 1.8:1 for 6i) by re-integrating the ¹H & ¹⁹F NMR spectra of corresponding compounds in the revised manuscript and SI. We have also carefully re-checked the spectral data of all the compounds and made corrections wherever necessary (the integration of peaks, coupling constants in the compound characterization, and their respective NMR spectra).

16. In the SI, some of the final compounds (3d, 3e, 3k, 3o, 3ae, 3af, 4r, 4x, 4ac and 5g etc.) still show solvent peaks or impurities in the spectra. The authors must provide pure spectra in the revision.

>> We have reacquired and replaced the NMR spectra of the above-mentioned compounds: 3d, 3e, 3k, 3o, 3ae (3ah), 3af (3ai), 4r (4s), 4x (4y), 4ac (4ad) and 5g. In addition to these, we have also changed the spectra of the following compounds: 3m, 4ad (4ae), 4ag (4ah), 5a, 5d, 5e, 6d, 6i, 7a, 8b (8g), 9 and 23 [in bracket, new numbering].

Reviewer #3

The authors have addressed all my concerns, regarding mechanistic study, substrates scope and other queries in the response letter with corresponding experimental results. These newly added results into the manuscript and supplementary information could provide a better understanding of this advanced methodology. As I mentioned in the previous comments, I won't agree on the true novelty of this methodology compared to existing literature data (*Tetrahedron Letters*, 1987, 28, 117-120; *Org. Lett.* 2023, 25, 3564–3567; *American Chemical Journal*, 1901, 26, 353; *Org. Biomol. Chem.*, 2008, 6, 2995–2999; *J. Am. Chem. Soc.*, 1988, 10, 8679-8685; *J. Serb. Chem. Soc.*, 2015, 80, 453–458; *J. Org. Chem.*, 1993, 58, 3355-3360). However, based on the authors exceptional additional experimental results, selectivity, broadness proved their hypothesis shows an advancement compared to existing literature data. Therefore, I would like to recommend this work in “nature communications” after major revision noted below.

>> We thank the reviewer for the supportive comment.

Detailed comments:

1. The authors must expand the scope of **2o** and diverse alkenes derivatives to showcase the broadness of this synthetic methodology similar to Scheme 3 and 4.

>> As suggested, we have further expanded the scope of **2o** (selenocyanatomalonate) with diverse alkene derivatives to showcase the broadness of this synthetic method. As shown below, the methodology worked well with various alkenes and selenocyanatomalonates to obtain corresponding *Se*- and *N*-heterocycles with moderate to good yield (44-72%) under the optimized conditions.

The results have been added to the lower part of Table 2 and have been discussed in the manuscript as well. The related spectra have also been added in SI (page no: S281-S285; S288-S292).

2. Why do the authors pick **2k** as a radical precursor instead of **2a** in Scheme 6 mechanistic studies? Because their main concept is to build products **3** and **4**.

>> We thank the reviewer for asking us this question. While we initially did start the mechanistic investigations with substrate **2a** (see Fig. 6B), we encountered problems with the isolation of compound **3a'** (as mentioned, it is metastable and cyclizes upon silica gel column chromatography), due to which we were unable to perform a controlled mechanistic study showing its conversion to **4a** (Fig. 6E-G, 6I and 6J). Moreover, the reaction between **1a** and **2a** under standard conditions provides uncyclized **3a'** on the one hand and the cyclized **4a** on the other, due to the presence of the enolizable hydrogen on **2a**, which makes any comparative analysis very difficult. To overcome this issue and make the mechanistic studies more streamlined, we worked with the quaternary alkyl thiocyanate **2k**, which does not have enolizable hydrogen and performed the required mechanistic investigations using its corresponding thiocyanate product **5f** and isothiocyanate product **6f**.

3. In this work, the authors synthesized thiocyanatomalonates from the corresponding malonates by a one-pot

process involving bromination, followed by thiocyanation. I will suggest trying the reaction by using diethyl 2-bromomalonate and ammonium thiocyanate with alkene under standard conditions (For example; Org. Lett. 2023, 25, 3564–3567). If this works, it will be interesting and improve the novelty of the reaction.

>> We thank the reviewer for the kind input. Indeed a three-component reaction with 4-methylstyrene **2a**, 2-bromomalonate, and ammonium thiocyanate does provide the desired chemo-divergent products **3a** and **4a** albeit under a slightly modified solvent system (toluene/MeCN, 1:1). However, the yield is only moderate to good (57–63%) and comparatively lower than our developed two-component bi-functional thiocyanomalonate chemistry. We have added this in Fig. 5A, along with a brief discussion in the manuscript: “*Interestingly, the reaction of 4-methylstyrene 1a with diethyl bromomalonate and ammonium thiocyanate under slightly modified conditions also delivered the desired chemo-divergent products 3a (63%) and 4a (57%), albeit less yield compared to the corresponding bifunctional thiocyanate 2a. This modular three-component recipe further enriches this developed method to access hydrothiophenes and pyrrolidine heterocycles from ready-stock materials by just regulating the photocatalyst (Fig. 5A).*”

4. Few compound HRMS analysis is not within the 10-ppm acceptable error (for example: **3a**, **3c**, **3t**, **3z** etc.). Please recheck in the entire SI and provide the original HRMS spectra for review.

>> We thank the reviewer. However, to the contrary, as per the guidelines of the journal, the acceptable range of the HRMS analysis is within 0.003 (i.e., 30 ppm) for molecules with $m/z < 1,000$. After carefully rechecking the HRMS spectra of all the compounds, we reacquired HRMS data of **3c**, **3d**, **3h**, and **3o**. These were previously beyond the 30 ppm limit but were found to be within the acceptable range upon reacquiring. We have also attached the original HRMS spectra of the mentioned compounds **3a**, **3c**, **3t** (**3u**), and **3z** (**3ac**), as well as the reacquired HRMS spectra in a separate file for review.

Mass spectrometry

Authors should also provide mass spectral data to support molecular weight identity. High-resolution mass spectral (HRMS) data are preferred, and when $m/z < 1,000$, the calculated and found values should be within 0.003. When $m/z > 1000$, we expect an experimental value within 1 ppm of the calculated value.

5. The authors showing compound **3ah**, 5i dr ratio is 1:1 but the respective NMR spectra are not in 1:1 ratio. On what basis the authors given this ratio? It is completely misleading the readers.

>> We thank the reviewer for bringing this to our notice. The dr ratio of **5i** is now modified with the exact NMR (^1H and ^{19}F) integration value (1.2:1). For **3ah** (**3ak**), the dr ratio is 1:1, including a minor fraction of rotamers.

6. Still some NMR spectra of compounds are not pure such as **4ad**, **4ag**, **5a**, **5d**, **5e**, **6d**, **6i**, **7a**, **8b** and **9**.

>> We have replaced the NMR spectra of following compounds: **4ad (4ae)**, **4ag (4ah)**, **5a**, **5d**, **5e**, **6d**, **6i**, **7a**, **8b (8g)**, and **9**. In addition to these, we have also re-acquired the spectra for compounds: **3d**, **3e**, **3k**, **3m**, **4o**, **3ae (3ah)**, **3af (3ai)**, **4r (4s)**, **4x (4y)**, **4ac (4ad)**, **5g**, and **23**. [in bracket, new numbering of compounds].

7. Still there are many typos and grammatical errors in the manuscript and supporting information. Few things are noticed below.

>> We thank the reviewer for the comment. We have thoroughly checked the manuscript and SI to rectify typos and grammatical errors.

a) In page 10, the color coating of the compound 9 is different with SI. Please maintain consistency.

>> We have carefully checked and maintained the consistency of the color coating of all the compounds in the revised manuscript and SI.

b) The “mol %” should be revised to “mol%” in the manuscript and SI where it is necessary.

>> We corrected the issue in the revised manuscript and SI.

c) Please write the respective solvent quantity (page S21; Solvent (x M)).

>> We have corrected it by replacing solvent concentration with solvent quantity in Supplementary Table 1 (page S21).

REVIEWERS' COMMENTS

Reviewer #2 (Remarks to the Author):

The authors once again have revised their manuscript according to the comments of all reviewers (mine and reviewer-3). Most importantly, inconsistency issues in the manuscript and SI were carefully solved. Some minor things do require further careful attention before the manuscript is ready for publication.

Comments:

1. Include my entire original introduction comments in the peer review file.
2. Please consider the revision of manuscript title because the whole concept is presenting current title (My suggestion; Photocascade Chemoselective Controlling of Ambident Thio(Seleno)cyanates with Alkenes via Catalyst Selective Electron Transfer).
3. In Fig. 1D, the R2 bond angle of alkene is incorrect. Please correct it.
4. According to TLC pictures of entry 2 in Table 1 of manuscript, the yields are does not matched; there is no 4a formation in TLC but authors given 7% yield. For better understanding include TLC pictures with and without iodine vapor.
5. I strongly suggest them to added all the failed substrate or incompatible substrates below the manuscript Schemes.
6. Please correct from "35-38%" to "35%-38%" in the entire manuscript where it is necessary.
7. Please include the dr or E/Z ratio in the proton NMR spectra of the products.

REVIEWERS' COMMENTS

Reviewer #2 (Remarks to the Author):

The authors once again have revised their manuscript according to the comments of all reviewers (mine and reviewer-3). Most importantly, inconsistency issues in the manuscript and SI were carefully solved. Some minor things do require further careful attention before the manuscript is ready for publication.

Comments:

1. Include my entire original introduction comments in the peer review file.

>> As suggested, we have included the original introduction comments of the previous round of peer review in a separate file.

2. Please consider the revision of manuscript title because the whole concept is presenting current title (My suggestion; Photocascade Chemoselective Controlling of Ambident Thio(Seleno)cyanates with Alkenes via Catalyst Selective Electron Transfer).

>> As suggested, we have modified the manuscript title accordingly.

3. In Fig. 1D, the R2 bond angle of alkene is incorrect. Please correct it.

>> We redrew the structure with the correct bond angle.

4. According to TLC pictures of entry 2 in Table 1 of manuscript, the yields are does not matched; there is no 4a formation in TLC but authors given 7% yield. For better understanding include TLC pictures with and without iodine vapor.

>> As suggested, we included the TLC picture with and without iodine vapor in the Supporting information (page S21).

5. I strongly suggest them to added all the failed substrate or incompatible substrates below the manuscript Schemes.

>> We thank the reviewer for the suggestion. However, we feel adding each structure to the end of every scheme is not necessary as most of these incompatible substrates are commonly known by their names and have been appropriately mentioned in the manuscript, additionally directing the readers to "See incompatible substrates list, Supplementary Table 6" at each point where such substrates are named, asking them to go through the structures if they desire to do so. Moreover, adding a few discrete structures at the end of each scheme when they have already been designed in a compact manner will leave unused spaces in each of the figures (Fig. 3, 4 and Table 2) that will not look good for aesthetic purposes. Additionally, presenting these structures in a single table (Supplementary Table 6) is convenient as well as easy for the readers to find all of these substrates and the reasons for their incompatibility at a single place.

6. Please correct from "35-38%" to "35%-38%" in the entire manuscript where it is necessary.

>> We corrected the presentation of range of yields in %.

7. Please include the dr or E/Z ratio in the proton NMR spectra of the products.

>> As suggested, we have included the dr or E/Z ratio in the ¹H-NMR spectra of the products in SI.